# Immune-mediated hookworm clearance and survival of a marine mammal decrease with warmer ocean temperatures

Mauricio Seguel[1][†]*, Felipe Montalva[2], Diego Perez-Venegas[3], Josefina Gutiérrez[4,5], Hector J Paves[6], Ananda Müller[7], Carola Valencia-Soto[4][‡], Elizabeth Howerth[1], Victoria Mendiola[1], Nicole Gottdenker[1]

[1]Department of Pathology, College of Veterinary Medicine, University of Georgia, Athens, United States; [2]Facultad de Ciencias Biologicas, Pontificia Universidad Catolica de Chile, Santiago, Chile; [3]PhD Program in Conservation Medicine, Facultad de Ecología y Recursos Naturales, Universidad Andrés Bello, Santiago, Chile; [4]Programa de Investigación Aplicada en Fauna Silvestre, Facultad de Ciencias Veterinarias, Universidad Austral de Chile, Valdivia, Chile; [5]Facultad de Ciencias Veterinarias, Instituto de Patología Animal, Universidad Austral de Chile, Valdivia, Chile; [6]Departamento de Ciencias Básicas, Universidad Santo Tomas, Osorno, Chile; [7]Instituto de Ciencias Clínicas Veterinarias, Universidad Austral de Chile, Valdivia, Chile

*For correspondence:
mseguel@uga.edu

Present address: [†]Odum School of Ecology, University of Georgia, Georgia, United States; [‡]Facultad de Medicina Veterinaria, Universidad San Sebastián, Santiago, Chile

Competing interests: The authors declare that no competing interests exist.

**Abstract** Increases in ocean temperature are associated with changes in the distribution of fish stocks, and the foraging regimes and maternal attendance patterns of marine mammals. However, it is not well understood how these changes affect offspring health and survival. The maternal attendance patterns and immunity of South American fur seals were assessed in a rookery where hookworm disease is the main cause of pup mortality. Pups receiving higher levels of maternal attendance had a positive energy balance and a more reactive immune system. These pups were able to expel hookworms through a specific immune mediated mechanism and survived the infection. Maternal attendance was higher in years with low sea surface temperature, therefore, the mean hookworm burden and mortality increased with sea surface temperature over a 10-year period. We provide a mechanistic explanation regarding how changes in ocean temperature and maternal care affect infectious diseases dynamics in a marine mammal.
DOI: https://doi.org/10.7554/eLife.38432.001

## Introduction

Marine mammals are a diverse group of top predators highly sensitive to changes in aquatic ecosystems (*Constable et al., 2014*). Within this group, fur seals and sea lions (otariids) breed and give birth on land but forage at sea, alternating periods of foraging in the ocean with periods of offspring attendance and nursing on land (income breeders) (*Stephens et al., 2009*). Therefore, otariids, like other marine mammals, are highly sensitive to local changes in prey distribution and abundance (*Trillmich et al., 1991*, *Constable et al., 2014*, *Elorriaga-Verplancken et al., 2016*). One of the most important indexes of the abundance of marine mammal prey is sea surface temperature (SST) (*Soto et al., 2006*, *Elorriaga-Verplancken et al., 2016*). Warmer SST indicates reduced nutrient upwelling, which is associated with reduced primary productivity and abundance of mesopelagic

**eLife digest** Every year off the coasts of Chile, Guafo Island becomes a nursery for South American fur seals pups. Mother fur seals leave their young on the beaches, going out at sea to hunt for fish before returning to the shore to nurse. These first few months are dangerous for young seals, with many dying because of hookworms, parasites that latch to the wall of the bowels to suck blood. However, the immune system of the pups is usually able to mount a response and fight off these parasites.

Even though the pups stay on land, their lives depend on the health of the ocean that feeds their nursing mothers. In recent years, sea temperature has been rising rapidly, which modifies winds and water currents. This can set off a chain of events that results in fewer fish being available for seals and other marine mammals to eat. Researchers know that years with warmer waters are associated with changes in the pattern of the mothers' hunting trips, more pups' deaths, and a weaker immune system in young fur seals. However, the mechanisms that connect these different factors are still unclear.

To explore this, Seguel et al. followed South American fur seals colonies on Guafo Island for several years, tracking the mothers' trips and monitoring the health of the pups by looking at their levels of blood sugar, whether they carry hookworms, and certain elements of their immune system. Results showed that in years when the sea is warmer, fur seal mothers are gone hunting for longer: they spend less time nursing their young, which then grow slower. These young seals also have lower levels of blood sugar, and so they have less energy to create the immune response necessary to clear off parasitic worms. In fact, in years with warmer seas, almost half of the pups die from hookworm infections.

The work by Seguel et al. shows that warmer oceans directly weaken the immune defenses of certain marine mammals. If temperatures keep rising, infectious diseases may kill more of these animals. Further work is now needed to explore if strategies could be developed to help seal populations, for example by treating the pups with drugs that eliminate the parasites.
DOI: https://doi.org/10.7554/eLife.38432.002

marine organisms (*Lewandowska et al., 2014*). This decrease in food resources forces otariid females to change their foraging strategies by increasing their foraging trip lengths, resulting in decreased time spent on land with their pup (maternal attendance) (*Trillmich et al., 1991*, *Costa, 2008*). These changes in patterns of maternal attendance have been associated with decreased pup growth and increased mortality (*Soto et al., 2006*, *Jeanniard-du-Dot et al., 2017*). Regardless, the mechanisms that drive decreased survival during years with low ocean productivity have not been intensely explored beyond assuming that this results from direct mortality because of starvation. However, in some otariid populations, in years with abnormal SST, the immune competence of pups decreases (*Banuet-Martínez et al., 2017*), suggesting that environmental variables can affect the health of marine mammals by impairing their immune function. If these immunological changes impact offspring survival, there could be additional negative consequences between a warmer ocean, health, and survival of marine vertebrates.

In some marine mammal populations, infectious diseases are one of the most significant causes of mortality among young individuals (*Gulland and Hall, 2007*; *Spraker et al., 2007*; *Seguel et al., 2013*). In otariids, hookworms (*Uncinaria sp.*) have been described in nearly all species, and while some populations suffer few adverse effects, others experience up to 70% of hookworm-related mortality being one of the most significant infectious diseases of young fur seals and sea lions (*Spraker et al., 2007*; *Lyons et al., 2011a*; *Seguel et al., 2013*; *Seguel and Gottdenker, 2017*). Fur seals are infected with hookworms (*Uncinaria sp.*) during their first 1–4 days of life through their mother's colostrum (*Lyons et al., 2011b*; *Seguel et al., 2018*). These nematodes live in the small intestine where they bite the mucosa to feed on blood, causing substantial tissue damage, anemia, and death (*Marcus et al., 2015*, *Seguel et al., 2017*, *Seguel et al., 2018*); however, it is unclear how the host responds to this infection. Long term studies in fur seal populations show that hookworm prevalence and mortality varies over time, but the mechanisms driving these patterns are unknown (*Lyons et al., 2011a*; *Seguel et al., 2013*). In this paper, we describe how oceanographic

environmental variables, via the modification of maternal care, are associated with immune-mediated parasite clearance, and survival of a marine mammal, the South American fur seal (SAFS, *Arctocephalus australis*).

## Results

### Hookworm disease dynamics and mortality in fur seal pups

The hookworm (*Uncinaria sp.*) prepatent period varied from 14 to 18 days and based on the coprological analyses and necropsies of recaptured pups, the number of days a pup released hookworm eggs (infectious period) ranged from 5 to 55 days (2014–15 and 2017, mean = 25.7 ± 10.9, n = 146). Seven to 15 days before having a negative coprological test, fur seal pups experienced a decline of more than 50% in the number of eggs shed in previous exams. At this stage, pups were considered to be in a hookworm clearance state. When presenting the first negative coprological exam, they were considered to have cleared hookworm infection (*Figure 1A*). Between 81% to 100% of pups examined through necropsy between 2005–08 (n = 124) and 2012–17 (n = 154) had evidence of hookworm infection, and hookworm-related mortality corresponded to 13–50% of all pups found dead (n = 56, *Figure 1B*). Total hookworm mortality could be calculated in a subset of marked pups

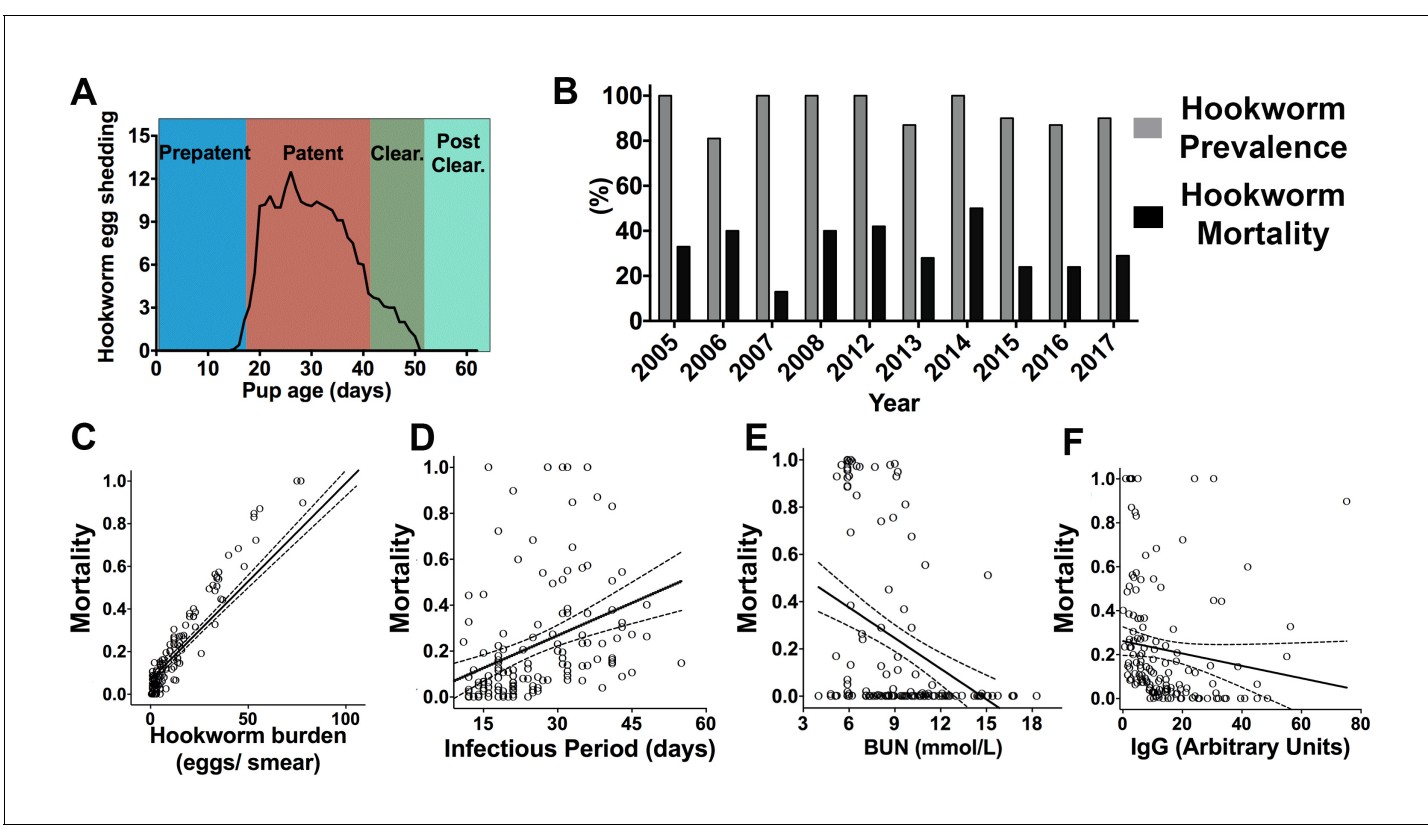

**Figure 1.** Hookworm infection dynamics and predictors of hookworm mortality in South American fur seal (*Arctocephalus australis*) pups at Guafo Island, southern Chile. (A) Hookworm egg shedding patterns through infection stages. (B) Hookworm prevalence and mortality through different reproductive seasons (2005–08, 2012–17). (C–F) Predictors of hookworm mortality in generalized linear mixed models (GLMM) (2014, 2015, 2017) vs observed hookworm mortality. Higher hookworm burdens (GLMM, 1.47 ± 0.56, Z = 2.61, p = 0.009), longer infectious periods (GLMM, 0.17 ± 0.06, Z = 2.87, p = 0.004), and lower plasma concentrations of blood urea nitrogen (BUN) (GLMM, −0.45 ± 0.23, Z = −1.97, p = 0.049) and parasite-specific IgG (GLMM, −0.41 ± 0.17, Z = −2.45, p = 0.014) increased the probability of hookworm mortality. Raw data in: *Figure 1—source data 1*.
DOI: https://doi.org/10.7554/eLife.38432.003

The following source data is available for figure 1:

**Source data 1.** Predictors of hookworm-related mortality in South American fur seal pups.
DOI: https://doi.org/10.7554/eLife.38432.004

in 2014 (n = 38), 2015 (n = 53), and 2017 (n = 54) (*Figure 1—source data 1*). Hookworms killed 42.1% of pups born in 2014, 20.7% of pups born in 2015, and 24% of pups born in 2017 at Guafo Island (GLM, 2014 = 1.02 ± 0.47, Z = 2.16, p = 0.0304). Based on multimodel inference using generalized linear mixed models, pups that had higher hookworm burden, delayed hookworm clearance, and lower plasma concentration of parasite specific IgG, blood urea nitrogen, and glucose were more likely to die from hookworm disease (*Figure 1C–F*) (*Supplementary file 1* and *2*). Therefore, the most important host-related factors affecting hookworm mortality were energy balance and immune response against the parasite. The parasite-related factors affecting mortality suggested that hookworm clearance, by reducing infectious period and hookworm burden, enhanced host survival.

## Hookworm clearance is immune-mediated

To determine the mechanisms that drive hookworm clearance and affect host mortality, the immune response to hookworms was investigated during 2017 at different infection stages in 54 fur seal pups, and compared to 24 hookworm-free (ivermectin-treated) age-matched controls (*Figures 2*, 4A and B, *Figure 2—source data 1*). The number of peripheral blood leukocytes (lymphocytes, macrophages, neutrophils, eosinophils, and basophils) was obtained as a basic tool to indirectly measure the level of proliferation of these different immune cell types in infected and control animals. During the patent and clearance period, fur seal pups that survived infection (n = 41) experienced a significant increase in the number of peripheral blood lymphocytes (GLMM, 0.9 ± 0.003, Z = 231, p = $2.0 \times 10^{-16}$) and basophils (GLMM, 4.8 ± 0.08, Z = 56.7, p = $2.0 \times 10^{-16}$), and had higher numbers of these cells when compared to age-matched controls and to the pups that died from hookworm infection (*Figure 2A–B*). The number of neutrophils in peripheral blood was similar between controls and pups that survived but slightly lower in pups that died (n = 13) from hookworm disease (GLMM, died = −0.53 ± 0.06, Z = −8.04, p = $9.1 \times 10^{-16}$). During the patent period, lower numbers of monocytes were found in animals that died from hookworm disease compared to controls (GLMM, died = −0.88 ± 0.11, Z = −7.64, p = $1.54 \times 10^{-14}$), and eosinophils were higher in animals that survived when compared to controls and animals that died (GLMM, survived = 0.86 ± 0.16, Z = 5.19, p = $2.0 \times 10^{-7}$); however, during the clearance and post-clearance periods, eosinophils (GLMM, survived = 0.26 ± 0.14, Z = 1.88, p = 0.07) and macrophages (GLMM, survived = −0.03 ± 0.09, Z = 0.36, p = 0.71) were in similar numbers in pups that survived infection and controls. Pups that cleared the infection developed medium to high levels of parasite-specific IgG, whereas the level of these antibodies was significantly lower in pups that died from hookworm infection and almost non-existent in the control group (*Figure 2J–M*). There was moderate to marked immunolabelling of the hookworm intestinal brush border using serum from six pups with moderate to high levels of parasite-specific IgG (23–100 arbitrary units) (*Figure 2I*), suggesting that anti-hookworm antibodies bind proteins located in the hookworm intestine.

To determine the morphological and immune cell population changes in the anatomical site of hookworm infection, sections of small intestine and mesenteric lymph nodes were collected from pups that died from hookworm disease (n = 21), pups that were undergoing clearance (n = 18), and pups that were never infected with hookworms (controls, n = 6) (*Figure 3—source data 1*). The small intestine mucosa, submucosa, and the mesenteric lymph nodes of pups undergoing hookworm clearance contained larger numbers of T-lymphocytes when compared to pups that died from hookworm infection or pups never infected with adult *Uncinaria sp* (Generalized linear models with negative binomial distribution (GLM.NB), mucosa clearance = 0.86 ± 0.11, Z = 8.312, p = $2.0 \times 10^{-16}$, submucosa clearance = 1.08 ± 0.16, Z = 6.86, p = $7.0 \times 10^{-12}$, mesenteric lymph node clearance = 0.78 ± 0.06, Z = 14.21, p = $2.0 \times 10^{-16}$) (*Figure 3*). B-lymphocytes and plasma cells were more numerous in the mesenteric lymph node of pups clearing hookworm infection versus controls and pups dead from hookworm infection (GLM.NB, B-lymphocytes clearance = 0.29 ± 0.07, Z = 4.1, p = $4.1 \times 10^{-5}$, plasma cell clearance = 0.59 ± 0.05, Z = 10.2, p = $2.0 \times 10^{-16}$). Similarly, there were higher numbers of mast cells (GLM.NB, clearance = 1.14 ± 0.25, Z = 4.6, p = $4.2 \times 10^{-6}$) and more mucus (GLM, clearance = 0.03 ± 0.003, Z = 7.84, p = $9.4 \times 10^{-10}$) in the mucosa, and more leukocytes expressing IL-4 in the intestine (GLM.NB, clearance = 1.97 ± 0.15, Z = 12.72, p = $2.0 \times 10^{-16}$) and mesenteric lymph node (GLM.NB, clearance = 1.57 ± 0.11, Z = 14.63, p = $2.0 \times 10^{-16}$) of pups that cleared hookworm infection when compared to controls and pups with hookworm enteritis and bacteremia. Pups that died from hookworms, however, had larger numbers of macrophages in the

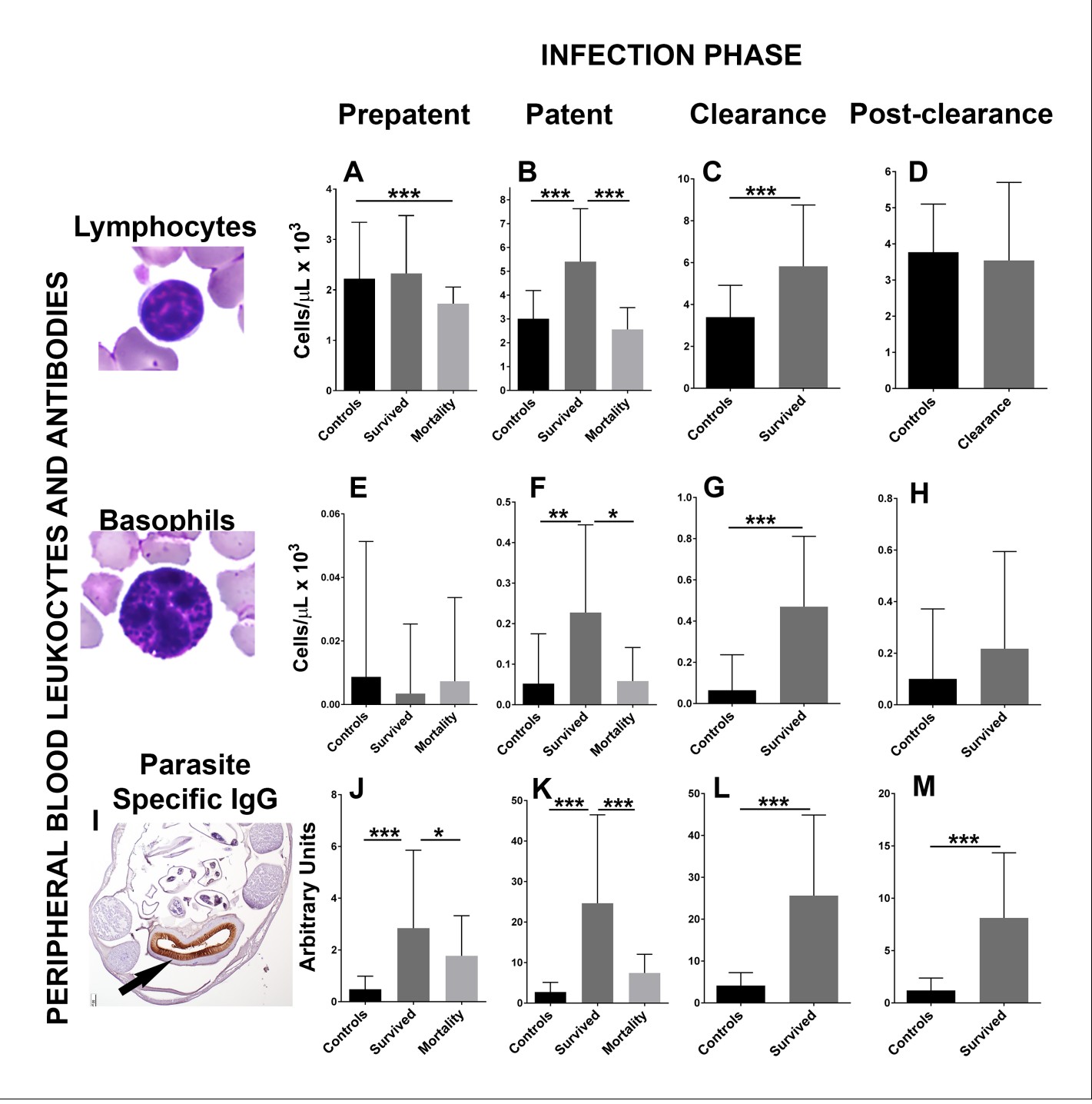

**Figure 2.** Changes in peripheral blood leukocytes and parasite-specific IgG antibodies during different phases of hookworm infection (*Uncinaria sp.*) in South American fur seal (*Arctocephalus australis*) pups (2017). (A) Pups that die from hookworm disease have lower numbers of lymphocytes during the prepatent phase compared to controls and pups that survived (Generalized linear mixed model (GLMM), lymphocytes died = $-0.52 \pm 0.13$, Z = $-4.06$, p = $4.92 \times 10^{-5}$). Pups that survive hookworm infection have higher numbers of lymphocytes during the patent (B) (GLMM, lymphocytes survived = $0.80 \pm 0.13$, Z = 6.04, p = $1.54 \times 10^{-9}$) and clearance (C) (GLMM, lymphocytes survived = $0.52 \pm 0.12$, Z = 4.30, p = $1.74 \times 10^{-5}$) infection phases when compared to pups that died due to hookworm infection and/or age matched controls. (E–H) Pups that clear and survive hookworm infection have markedly higher numbers of basophils during the patent (GLMM, basophils survived = $7.46 \pm 1.4$, Z = 5.33, p = $1.0 \times 10^{-7}$) and clearance (GLMM, basophils survived = $6.34 \pm 0.9$, Z = 7.07, p = $1.5 \times 10^{-12}$) infection phases compared to controls and pups that died from hookworm infection. (I) Fur seal pups that clear hookworm infection produce parasite-specific IgG that binds the intestinal brush border of the fur seal hookworms (*Uncinaria sp.*) (arrow). (J–M) Fur seal pups that clear hookworm infection have higher levels of parasite-specific IgG during the prepatent (GLMM, IgG

*Figure 2 continued on next page*

*Figure 2 continued*

survived = 1.78 ± 0.36, Z = 4.86, p = 1.12×10$^{-6}$), patent (GLMM, IgG survived = 2.27 ± 0.25, Z = 9.067, p = 2.0×10$^{-16}$), clearance (GLMM, IgG survived = 1.80 ± 0.2, Z = 9.0, p = 2.0×10$^{-16}$), and post-clearance (GLMM, IgG survived = 1.87 ± 0.25, Z = 7.3, p = 3.53×10$^{-13}$) infection phases compared to controls and pups that died. Asterisk indicate groups are statistically different at alpha = 0.05. p-values code: *0.01 < 0.05, **0.001 < 0.01, *** < 0.001. Raw data in: **Figure 2—source data 1**.

DOI: https://doi.org/10.7554/eLife.38432.005

The following source data is available for figure 2:

**Source data 1.** Maternal attendance and health-related parameters in pups with different hookworm infection status.

DOI: https://doi.org/10.7554/eLife.38432.006

intestinal submucosa (GLM.NB, mortality = 0.52 ± 0.09, Z = 5.79, p = 6.94×10$^{-9}$) and mesenteric lymph nodes (GLM.NB, mortality = 0.63 ± 0.05, Z = 12.68, p = 2.0×10$^{-16}$) compared to pups never infected with hookworms and pups clearing hookworm infection (**Figure 3**).

## Maternal attendance affects fur seal pup hookworm clearance

Maternal attendance patterns and pup-related health parameters were assessed in the 2017 reproductive season (n = 78) (**Figure 4A and B**, **Figure 4—source data 1**). Among measured serum chemistry variables, the average level of blood glucose was the best predictor of growth rate (GLM.NB, 0.18 ± 0.03, Z = 5.9, p = 2×10$^{-16}$). Among the considered external factors that could affect growth, the number of nursing events observed in a pup was the most significant predictor of growth rate, and although hookworm burden and hookworm infectious period were included in some top ranked models, their effect was not significant (**Figure 4A**, **supplementary file 4** and **5**). Additionally, there were no significant differences in growth rates between pups treated with ivermectin (n = 24) and non-treated (n = 54) (GLM.NB, 0.17 ± 0.15, Z = 1.16, p = 0.26). Nevertheless, when pups that died from hookworm disease were considered (n = 13), they had significantly slower growth rates (GLM.NB, −1.05 ± 0.16, Z = −6.6, p = 2.7×10$^{-11}$) compared to pups that survived (n = 41) and pups treated with ivermectin; however, the animals that died also had the lowest levels of maternal attendance (GLM.NB, −0.78 ± 0.23, Z = −3.45, p = 5.5×10$^{-4}$) (**Figure 4B**). Regarding the factors that affected overall immune reactivity (**Figure 4C**, **Figure 4—source data 1**), pups with more nursing events, faster growth rate, and higher hookworm burden were more likely to recruit higher numbers of T-cells (CD3+ lymphocytes) in the skin in response to (Phytohemagglutinin) PHA challenge (**Figure 4C**, **supplementary file 5** and **6**).

Pups with lower parasite-specific IgG concentrations (GLMM.NB, coeff = −0.017 ± 0.002, Z = 6.54, p = 2×10$^{-16}$, n = 146) and higher hookworm burden (GLMM.NB, coeff = 0.06 ± 0.022, Z = 2.78, p = 0.005, n = 146) had longer infectious periods (**Supplementary file 7** and **8**), suggesting that among measured immune parameters, parasite-specific IgG was the most significant factor affecting the permanence of hookworms in the intestine. Based on the PHA immune challenge performed when pups were 1-mo-old (**Figure 4D**, **Figure 4—source data 1**), animals with high T-cell response had higher levels of IgG, maternal attendance, glucose, growth rate, and shorter infectious periods at the end of the study when compared to the average levels in pups with low T-cell response (**Figure 4D**). However, hookworm burden was similar between the two groups (**Figure 4D**), suggesting, in conjunction with the previous analyses, that maternal attendance and growth rate accounted for most of the difference in T-cell reactivity between these groups.

## In years with high sea surface temperature there is lower maternal attendance, immune response, and increased hookworm-induced mortality

SAFS females were observed more frequently arriving to the rookery from foraging trips early in the morning (2007 = 78/115, 67.8% returning events in the morning, 2017 = 87/135, 64% returning events in the morning). Foraging trip length was correlated with the number of nursing events, indicating that the more time females spend at sea makes it less likely to observe them nursing their pup (**Figure 5A**). In 2017, a year with SST above Guafo Island average, SAFS females (n = 21) spend more time foraging at sea compared to 2007 (n = 23), a year with SST below Guafo Island average, therefore in 2017 (n = 79) the level of maternal attendance and pup growth rate were lower than in

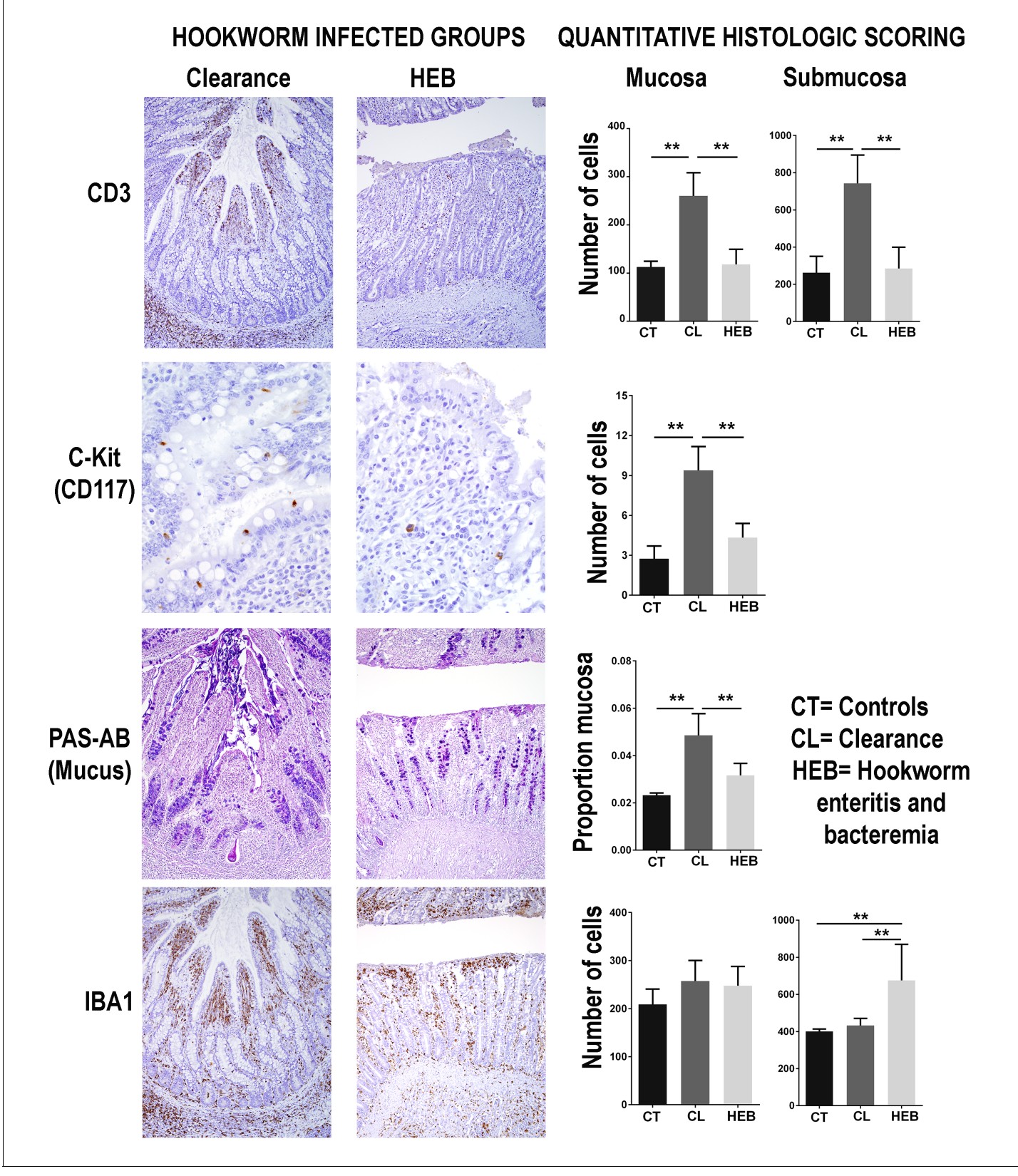

**Figure 3.** Intestinal immune response in different groups of South American fur seals (*Arctocephalus australis*) infected with hookworms (*Uncinaria sp.*) and controls. During the clearance process, fur seal pups recruit numerous T-lymphocytes (CD3 stain) in the jejunum mucosa (Generalized linear models

*Figure 3 continued on next page*

*Figure 3 continued*

with negative binomial distribution (GLM.NB), clearance = 0.86 ± 0.11, Z = 8.312, p = $2.0\times10^{-16}$) and submucosa (GLM.NB, clearance = 1.08 ± 0.16, Z = 6.86, p = $7.0\times10^{-12}$). Mast cells (C-kit stain) are found in higher numbers in the intestinal mucosa of pups undergoing clearance (GLM.NB, clearance = 1.14 ± 0.25, Z = 4.6, p = $4.2\times10^{-6}$). The intestinal mucosa of pups clearing hookworm infection contains a large amount of mucus (GLM, clearance = 0.03 ± 0.003, Z = 7.84, p = $9.4\times10^{-10}$). Pups that die from hookworm enteritis and bacteremia (HEB) have lower numbers or proportions of these immune components but higher numbers of macrophages (IBA1 stain) in the jejunum submucosa (GLM.NB, mortality = 0.52 ± 0.09, Z = 5.79, p = $6.94\times10^{-9}$). Asterisks indicate groups are statistically different at alpha = 0.05. p-values code: *0.01 < 0.05, ** <0.01. GLM.NB. Raw data: *Figure 3— source data 1*.

DOI: https://doi.org/10.7554/eLife.38432.007

The following source data is available for figure 3:

**Source data 1.** Leukocyte subsets and IL-4 immunohistochemical staining in intestine and mesenteric lymph node.

DOI: https://doi.org/10.7554/eLife.38432.008

2007 (n = 128) (*Figure 5B*) (*Figure 5A and B*, *Figure 5—source data 1*). Between 2012 and 2017 (*Figure 5C*, *Figure 5—source data 1*), in years with high SST (e.g. 2014), the average concentrations of glucose, cholesterol, parasite-specific IgG, and peripheral blood lymphocytes and basophils were lower than in years with low SST (*Figure 5C*). Similarly, the average hookworm infectious period was shorter in years with low SST (GLM, $X^2$ = 6.95, df = 1, p = 0.00036).

Over a 10-y period (2005–08, 2012–17) (*Figure 6*, *Figure 6—source data 1* and *2*), there was a significant positive correlation between mean hookworm burdens of necropsied pups and SST (Linear regression, Ad-$R^2$ = 0.86, p < 0.001), and between hookworm mortality and SST (Ad-$R^2$ = 0.56, p = 0.016); however, in the case of hookworm prevalence at necropsy the correlation with SST was not significant (*Figure 6*, *supplementary file 9–11*) (Ad-$R^2$ = 0.29, p = 0.064). A similar but negative correlation existed between the same hookworm epidemiological parameters and average chlorophyll-a concentrations (*Figure 6*).

## Discussion

Hookworm disease causes significant mortality in SAFSs in the Chilean Patagonia and represents one of the most significant causes of death among pups (*Seguel et al., 2013*; *Seguel et al., 2018*). However, in some years, hookworm-induced mortality decreases significantly. We showed that variations in hookworm disease morbidity and mortality are associated with specific changes in the immune response against the parasite. Additionally, maternal care was the most important external factor affecting immune response and hookworm clearance. As otariid maternal care is influenced by ocean environmental conditions and prey availability (*Trillmich et al., 1991*; *Soto et al., 2006*; *Jeanniard-du-Dot et al., 2017*), indirect indicators of ocean productivity such as sea surface temperature are correlated with hookworm disease dynamics and overall fur seal pup survival. Changes in ocean productivity have been indicated as the main environmental factor driving marine mammal mortality (*Trillmich et al., 1991*; *Soto et al., 2006*, *Costa, 2012*, *Elorriaga-Verplancken et al., 2016*); however, the links between indexes of ocean productivity such as SST and mortality are based on circumstantial evidence, and the mechanisms that drive this correlation are unclear. Our study provides a mechanistic explanation regarding the specific pathophysiologic processes affected by changes in the marine environment, and explains how these environmental processes impact host physiology, immune response, and survival (*Figure 7*).

Based on the epidemiological data collected in the current study when fur seal pups are on average 2.5 mo old, adult hookworms cannot be found in the fur seal population because they have been cleared from the intestine or have died along with their host. This is consistent with what we have found in necropsies over more than 10 y at Guafo Island and with what has been suggested in other fur seal species (*Lyons et al., 2011b*; *Seguel et al., 2018*). Early hookworm clearance is a unique feature of otariids, as all studied natural hookworm infections of land mammals consist of long-term intestinal infections (*Seguel and Gottdenker, 2017*), suggesting that fur seals have developed efficient mechanisms to clear hookworms. Based on field experiments and the immune parameters measured in the present study, this parasite clearance process is mediated mostly by T-lymphocytes, basophils, Th2-type leukocytes, and production of parasite-specific IgG. These results are similar to changes in peripheral blood leukocytes in humans and rodents infected with

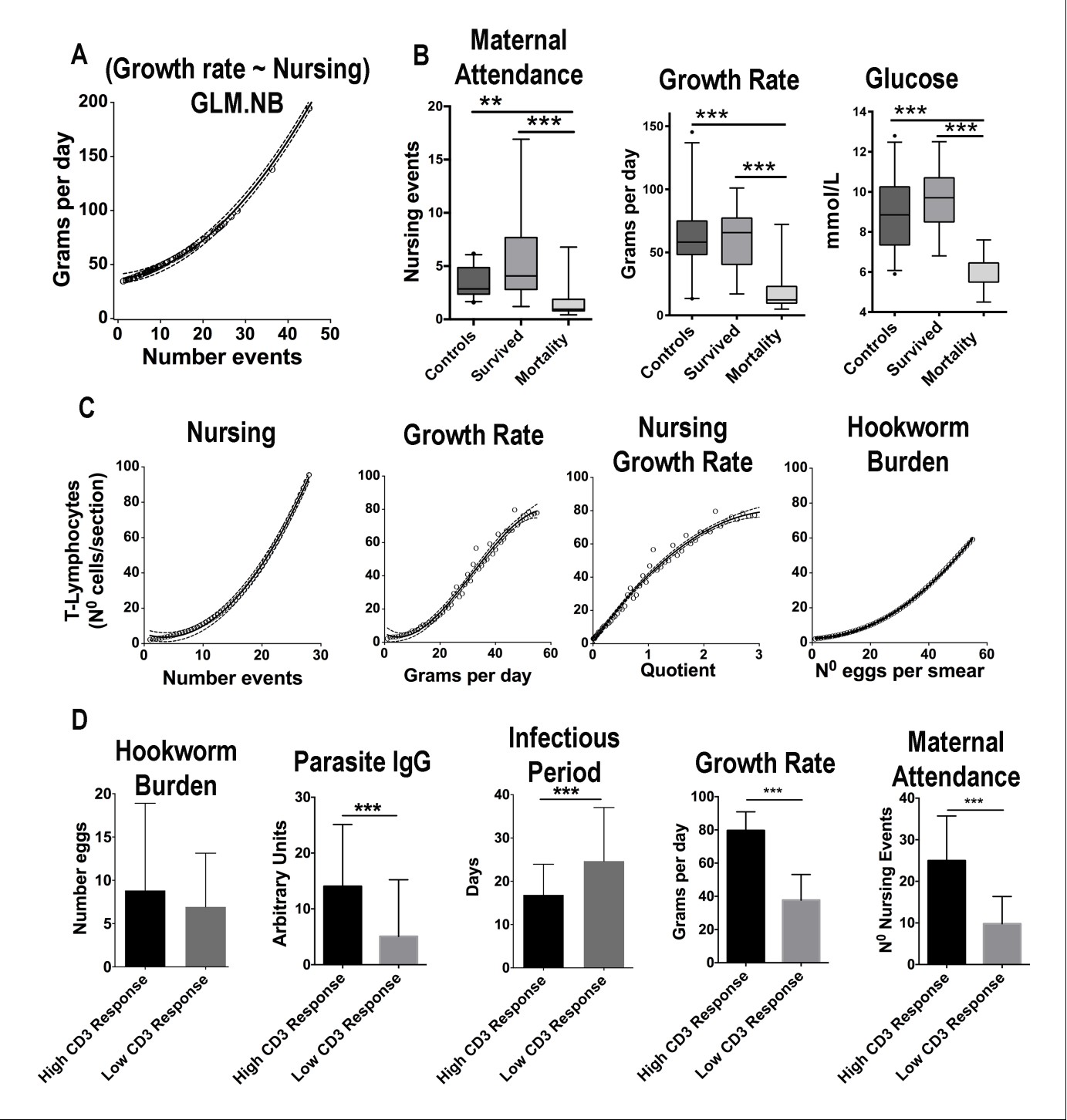

**Figure 4.** Maternal attendance affects South American fur seal pup's growth rate, energy balance, and immune response against hookworms. (**A**) The observed number of nursing events against predicted values of growth rate. With more nursing events pups grow faster (GLM.NB, 0.031 ± 0.006, Z = 5.53, p = 3.2×10⁻⁸). (**B**) Pups that survived hookworm infection had higher levels of maternal attendance (more nursing events) (GLM.NB, 0.78 ± 0.23, Z = 3.45, p = 5.5×10⁻⁴), faster growth rate (GLM.NB, 1.05 ± 0.16, Z = 6.6, p = 2.7×10⁻¹¹), and higher average levels of glucose (GLM, 3.0 ± 0.5, Z = 5.9, p = 1.02×10⁻⁷) compared to pups that died from hookworm disease; however, they had similar attendance and metabolic values compared to hookworm-free (control) pups (GLM.NB, 0.01 ± 0.13, Z = 0.13, p = 0.893, and GLM, 0.71 ± 0.38, t = 1.863, p = 0.066). (**C**) The observed values of number of nursing events, growth rate, interaction between nursing and growth rate and hookworm burden vs. the predicted values of CD3+ lymphocytes in section of skin in response to phytohemagglutinin (PHA) challenge. Pups with more nursing events (GLM.NB, 0.098 ± 0.02, Z = 4.39, p = 1.14×10⁻⁵), faster growth rate (GLM.NB, 0.04 ± 0.004, Z = 11.3, p = 2.0×10⁻¹⁶), and higher hookworm burden (GLM.NB, 0.009 ± 0.004,

*Figure 4 continued on next page*

*Figure 4 continued*

Z = 2.56, p = 0.01) had more recruitment of T-lymphocytes. (**D**) A subset of pups was divided into groups of low and high response to PHA challenge at 30 days old. Pups with higher CD3 lymphocyte response had higher average levels of parasite-specific IgG (GLM.NB, 1.11 ± 0.33, Z = 3.37, p = 7.5×10$^{-4}$), shorter infectious period (GLM.NB, −0.38 ± 0.12, Z = −3.06, p = 2.1×10$^{-3}$), faster growth rate (GLM.NB, 0.68 ± 0.06, Z = 10.9, p = 2.0×10$^{-16}$), and higher levels of maternal attendance (GLM.NB, 0.94 ± 0.13, Z = 7.04, p = 1.92×10$^{-1}$). Hookworm burden was similar between the two groups (GLM.NB, low reactivity = −0.63 ± 0.37, Z = −1.67, p = 0.09). Raw data: *Figure 4—source data 1* and *Figure 4—source data 2*.

DOI: https://doi.org/10.7554/eLife.38432.009

The following source data is available for figure 4:

**Source data 1.** Phytohemagglutinin immune challenge in 8-wk-old pups.
DOI: https://doi.org/10.7554/eLife.38432.010

**Source data 2.** Phytohemagglutinin immune challenge in 4-wk-old pups.
DOI: https://doi.org/10.7554/eLife.38432.011

hookworms (*Loukas and Prociv, 2001*; *Cortés et al., 2017*). In both systems, an increase in circulating T-lymphocytes and basophils is associated with hookworm clearance and resistance to re-infection (*Cortés et al., 2017*). Regarding parasite-specific IgG production, it is interesting that this antibody binds antigens located in a specific portion of the hookworm anatomy, the intestinal brush border. This anatomical location contains several digestive and heme-detoxifying enzymes that are crucial for the nematode blood digestion and survival (*Williamson et al., 2003*; *Wei et al., 2016*). In fur seals, it is possible that parasite-specific IgG reaches the nematode intestine with each blood meal, impairing the nematode blood digestion by blocking digestive enzymes, favoring clearance. A similar mechanism has been experimentally induced in dogs and humans through a hookworm vaccine using the digestive enzymes in the nematode intestinal brush border as antigens (*Hotez et al., 2016*; *Diemert et al., 2017*). These vaccines successfully avoid hookworm development and promote clearance (*Hotez et al., 2016*; *Diemert et al., 2017*). The morphologic and leukocyte population changes observed in the intestine of pups clearing hookworm infection are similar to those described in laboratory rodent models of hookworm infection. In these systems, T-lymphocytes, mast cells, and basophils are important players in the morphological changes of the intestine that facilitate nematode detachment and removal (*Ohnmacht and Voehringer, 2010*; *Cortés et al., 2017*). Therefore, as shown in laboratory animal models, it is likely that in fur seal pups, these changes in the intestinal mucosa and antibody production create a hostile environment for hookworm attachment and feeding, leading to clearance from the intestine.

Fur seal pups infected with hookworms for a shorter period of time were more likely to survive the infection. This suggests that there is a time-dependent effect of hookworms on the host, probably associated with host resource depletion, which could increase the risk of mortality from hookworm-induced anemia and peritonitis (*Spraker et al., 2007*; *Seguel et al., 2017*; *Seguel et al., 2018*). Additionally, pups with higher levels of parasite-specific IgG had shorter hookworm infection. This suggests that parasite-specific IgG is an immune element that contributes significantly to survival of fur seal pups. This is similar to what has been found in a wild population of Soay sheep (*Ovis aries*), where parasite-specific IgG is the best predictor of survival through the winter, precisely when there is scarcity of food resources and likely a more detrimental effect of parasites in the host (*Watson et al., 2016*). In most studied mammal species, IgG production is dependent on T-cell activity and caloric intake, particularly in neonates and children (*Papier et al., 2014*; *Ibrahim et al., 2017*). Similar processes could occur in fur seals given the differences observed in immune system reactivity and hookworm infection outcome in pups with different energy budget and maternal attendance patterns.

As expected, and as reported in other studies in pinnipeds, pups with higher levels of maternal attendance had higher growth rates (*Francis et al., 1998*; *Georges and Guinet, 2000*; *Arnould and Hindell, 2001a*). This could be related to pups spending more time with their mothers, shortening the fasting period. This is in line with the finding that in 2017 pups with more maternal attendance had higher glucose levels. Even though in some fur seal species pups can maintain relatively stable glucose levels during fasting (*Arnould and Hindell, 2001a*; *Verrier et al., 2012*), after a couple of days without nursing there is a significant decrease in blood glucose (*Arnould et al., 2001b*; *Champagne et al., 2012*). Therefore, it is not surprising that the average blood glucose levels and growth rate of pups were highly correlated in our study, suggesting that glucose could be a good

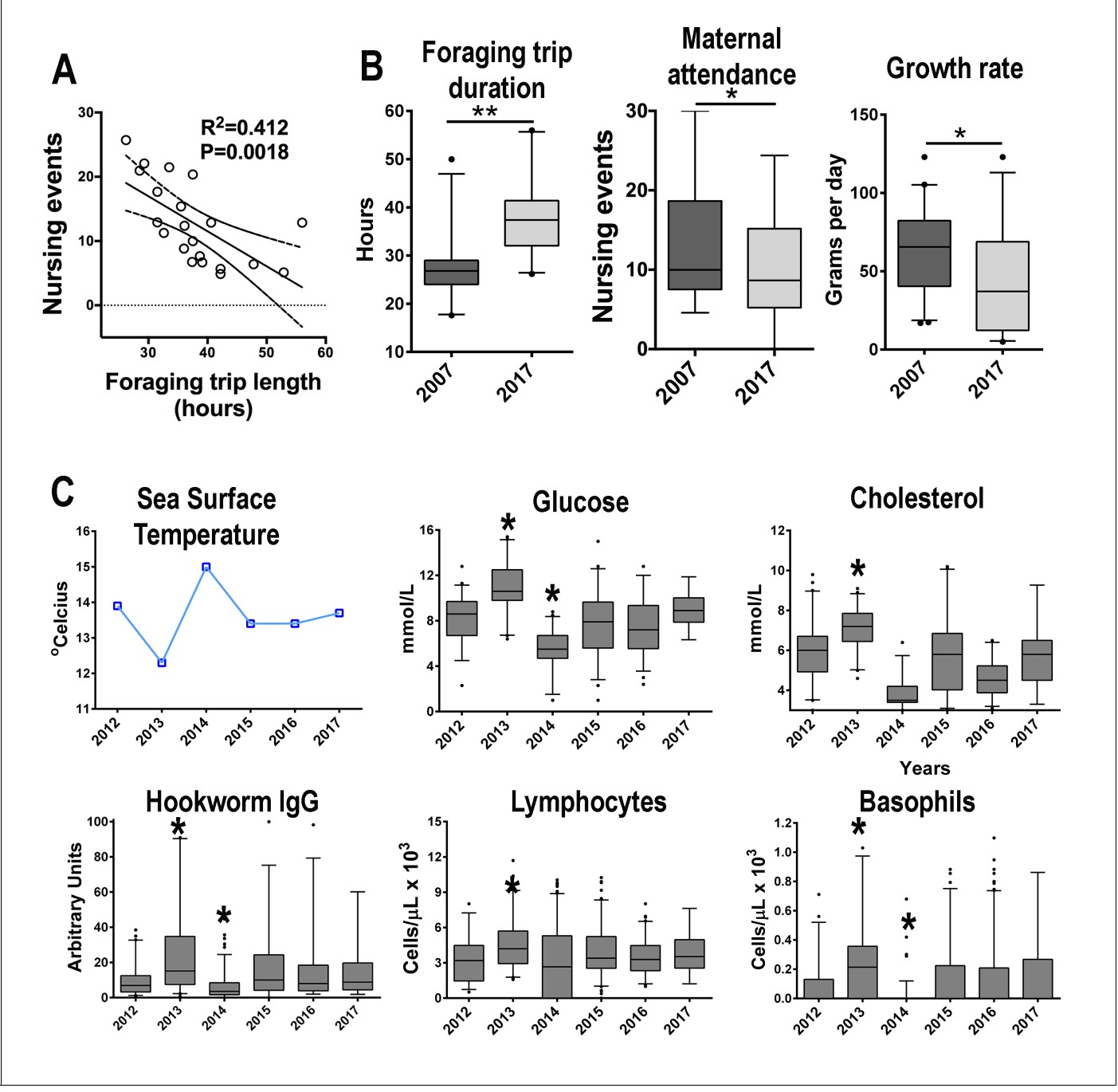

**Figure 5.** South American fur seals foraging behavior and maternal care patterns differ between seasons. (**A**) The level of maternal attendance decreases as foraging trips become longer (linear regression, $R^2$ = 0.412, p = 0.016; dashed lines represent 95% confidence intervals). (**B**) In a year with sea surface temperature (SST) below the historic Guafo Island average (2007), fur seal females foraging trips are shorter when compared to the mean foraging trip duration during a year with SST temperature above the historical average (2017) (unpaired T-test, t = 5.133, df = 42, p < 0.0001). Additionally, maternal attendance and pup growth rate in 2007 were higher than attendance and growth rates in 2017 (maternal attendance index: unpaired T-test, t = 2.060, df = 244, p = 0.04; growth rate: unpaired T-test, t = 2.85, df = 66, p = 0.0058). (**C**) Between 2012 and 2017 the mean values of glucose, cholesterol, parasite-specific IgG, peripheral blood lymphocytes and basophils followed an inverse pattern with mean SST at Guafo Island. In 2013, a year with low SST, pups had, on average, higher levels of glucose, cholesterol, parasite-specific IgG, lymphocytes and basophils when compared to the mean values of other reproductive seasons (Kruskal-Wallis with Dunn's multiple comparison tests, Kruskal-Wallis statistic = 73.2–114.6, mean rank diff. = 63.98–203.8, p < 0.0001–0.023). In 2014, with the highest mean SST over the last 15 y at Guafo Island, fur seal pups had the lowest mean values of these metabolic and immune parameters (Kruskal-Wallis with Dunn's multiple comparison tests, Kruskal-Wallis statistic = 73.2–114.6,

*Figure 5 continued on next page*

*Figure 5 continued*

mean rank diff. = −230.83,–83.4, p < 0.0001–0.017). (Asterisks indicate mean is significantly different from means of other seasons). Raw data in: **Figure 5—source data 1** and **Figure 5—source data 2**.

DOI: https://doi.org/10.7554/eLife.38432.012

The following source data is available for figure 5:

**Source data 1.** Maternal attendance and growth rates in 2007 and 2017.

DOI: https://doi.org/10.7554/eLife.38432.013

**Source data 2.** Immune and metabolic parameters in South American fur seal pups between 2012 and 2017.

DOI: https://doi.org/10.7554/eLife.38432.014

marker of the energy balance in a pup, and an important factor to consider in the health assessment of wild pinnipeds given the connection between energy balance and immune system. For instance, recent studies have shown that Californian sea lion pups with better body condition and higher glucose levels have higher total IgG (*Banuet-Martínez et al., 2017*). In humans and laboratory animals, glucose metabolism is one of the most important factors driving T-cell activity and production of antigen-specific IgG (*Mohammed et al., 2012*; *Palmer et al., 2015*). In our study, glucose levels and maternal attendance were the most significant factors affecting the overall level of T-cell reactivity in fur seal pups, and pups with higher T-lymphocyte reactivity produced higher levels of parasite-specific IgG and expelled hookworms faster. This suggests that fur seal pups that spend more time with their mothers receive more and/or higher quality milk, favoring a positive energy balance, leaving more energy available for immune response, including proliferation of T-lymphocytes and production of anti-hookworm antibodies.

Fur seal and sea lion maternal attendance patterns can be affected by several factors, including prey availability, maternal experience, and body condition (*Georges and Guinet, 2000*; *Arnould and Hindell, 2001a*; *Soto et al., 2006*; *McDonald et al., 2012a*; *McDonald et al., 2012b*). Out of these variables, body condition and prey availability can be affected by changes in sea surface temperatures (SST) because of the role that ocean temperature plays in nutrient upwelling, primary productivity, and distribution of fish stocks (*Soto et al., 2006*, *Costa, 2008*). In the southern Pacific Ocean, warmer ocean temperature is associated with a marked decrease in primary productivity and fish stocks, these effects being particularly intense during El Nino Southern Oscillation (ENSO) events (*Trillmich et al., 1991*; *Soto et al., 2006*). Additionally, in 'El Nino' years, the isotopic signature of pup tissues changes, which suggest shifts in the foraging and attendance regimes of their mothers (*Elorriaga-Verplancken et al., 2016*). In the Northern Chilean Patagonia, in years with higher SST, there is an apparent increase in the foraging trip length and a decrease in the maternal attendance of SAFS females. Because of the nature of our data acquisition, we cannot discard changes in nocturnal maternal attendance patterns, although the similarity in the proportion of foraging events recorded in the morning in years with high and low SST suggests that this is a less likely possibility. The reported changes in attendance coincide with slower growth rates of fur seal pups in a year with high SST compared to a year with low SST, suggesting that there could be a decrease in prey availability for SAFSs in years with higher SST, forcing them to spend more time foraging in the ocean and less time nursing their pups. This interpretation of the results is in line with the proposed conceptual model on the effect of environmental fluctuation in parental attendance in sea lions and fur seals (*Costa, 2008*). This model suggests that when environmental variation affects prey resources, adult females will increase their foraging intensity effort and metabolic rate before increasing foraging trip length, because the latter almost always results in a decrease in the net energy delivered to the offspring (*Costa, 2008*; *Costa, 2012*). In this context, the changes in maternal attendance observed in our study could explain the decreased average levels of glucose and cholesterol in fur seal pups in years with higher SST. Given the results of field experiments, it could be that in years with pup lower energy balance, immune parameters could also be decreased in these animals. This was observed, particularly for immune parameters such as circulating lymphocytes, basophils, and parasite-specific IgG, which are key elements for prompt hookworm clearance. Additionally, as these immune elements drive hookworm permanence in the pups' intestine (infectious period), in years with lower SST, hookworm infectious period was shorter, suggesting that environmental variables affect not only hookworm immune response but also transmission patterns.

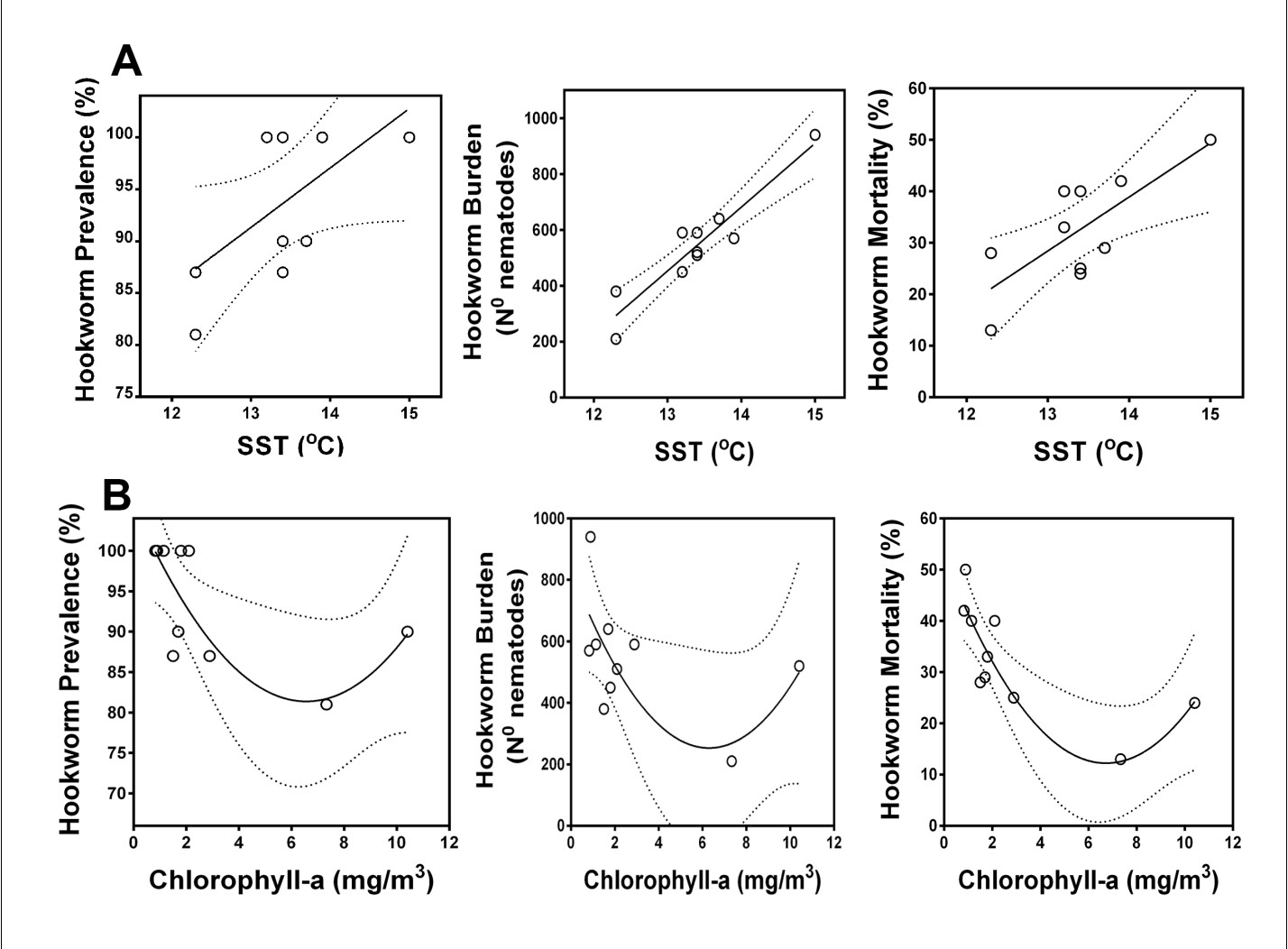

**Figure 6.** Correlation between oceanographic parameters (sea surface temperature and chlorophyll-a) and hookworm disease dynamics in South American fur seals (*Arctocephalus australis*) at the Chilean Patagonia (2005–08, 2012–17). (**A**) Hookworm prevalence, burden, and mortality increase in years with warmer sea surface temperature (Linear regressions. Hookworm prevalence, Ad-$R^2$ = 0.29, p = 0.064. Hookworm burden, Ad-$R^2$ = 0.86, p < 0.001. Hookworm mortality, Ad-$R^2$ = 0.56, p = 0.016). Hookworm prevalence, burden, and mortality decrease in some years with higher primary productivity (Second order polynomial regressions. Hookworm prevalence, Ad-$R^2$ = 0.46, p = 0.046. Hookworm burden, Ad-$R^2$ = 0.29, p < 0.123. Hookworm mortality, Ad-$R^2$ = 0.70, p = 0.005). Dashed lines represent 95% confidence intervals. Raw data: *Figure 6—source data 1* and *Figure 6—source data 2*.

DOI: https://doi.org/10.7554/eLife.38432.015

The following source data is available for figure 6:

**Source data 1.** Sea surface temperature data for Guafo Island.
DOI: https://doi.org/10.7554/eLife.38432.016
**Source data 2.** Hookworm prevalence, burden, and mortality in South American fur seal pups at Guafo Island.
DOI: https://doi.org/10.7554/eLife.38432.017

The present findings suggest that indirectly, SST modifies hookworm infection dynamics in SAFSs by impacting on maternal attendance patterns, energy budget between dams and pups, and pup immune response. This observation is also supported by the correlation found between SST and hookworm burden and mortality over a 10-y period on Guafo Island. Similar long-term studies on hookworm infection of fur seals have found dramatic changes in hookworm prevalence and mortality over a 20–30-y period (*Lyons et al., 2011b*). It is possible that changes in prevalence are associated with modification in host population density, given the sensitivity of this parasite to host density

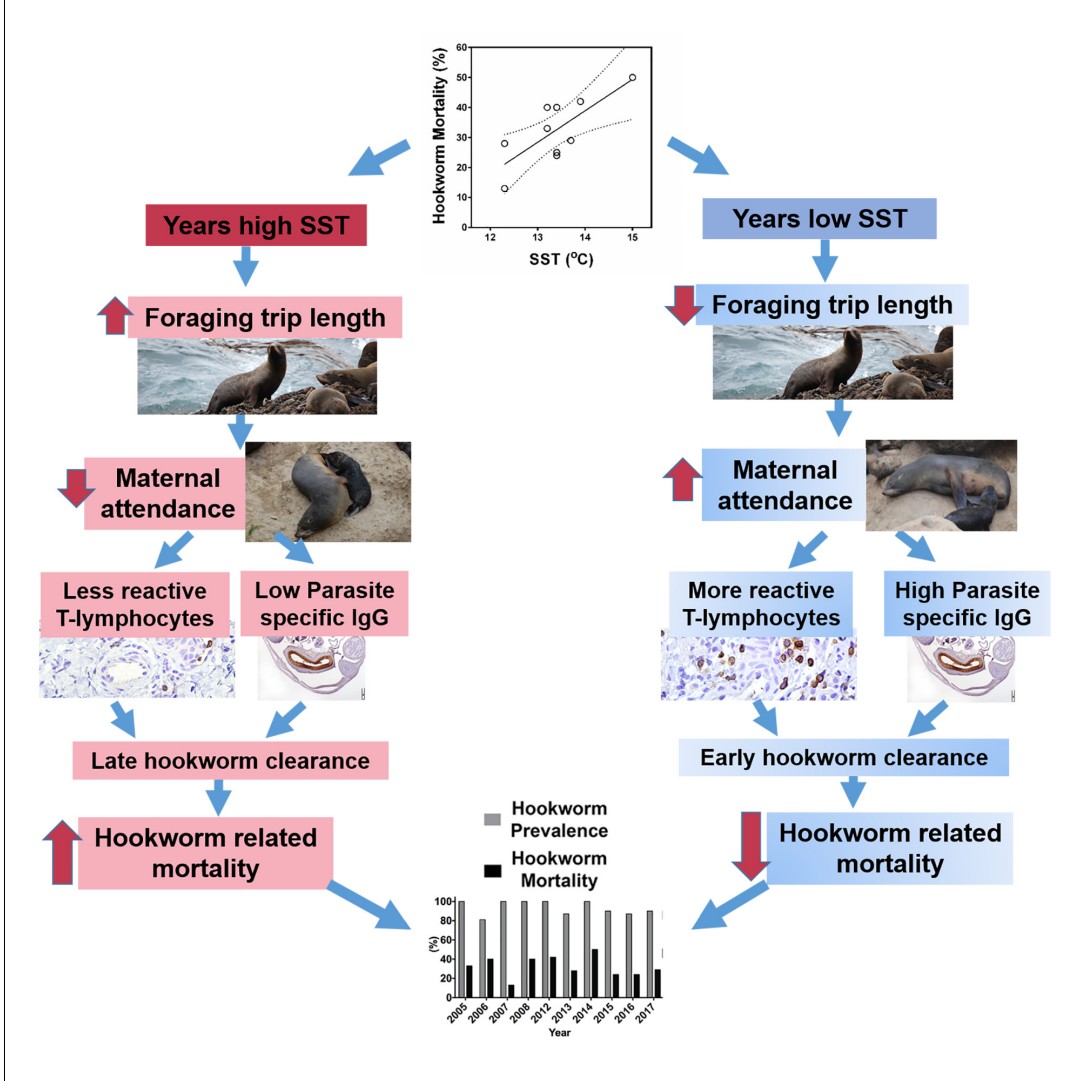

**Figure 7.** Proposed mechanism of South American fur seal response to environmental change in the context of endemic hookworm infection. Changes in sea surface temperature (SST) are associated with changes in foraging trip length and patterns of maternal attendance in fur seals. In years with lower SST, fur seal pups receive more diurnal maternal attendance compared to years with high SST. Pups with more maternal attendance have better energy balance, higher T-lymphocyte reactivity, and produce higher levels of parasite-specific IgG. Pups with this immune profile eliminate hookworms from the intestine faster than pups with less reactive immune systems. Early hookworm clearance is one of the most important factors that drive hookworm-related mortality, therefore in years with higher SST fur seal pups die more often of hookworm disease.
DOI: https://doi.org/10.7554/eLife.38432.018

constraint (*Lyons et al., 2011a*; *Seguel et al., 2018*). Our observations could explain why changes in prevalence and its correlation to environmental variables were not significant in our study, as the Guafo Island population has remained stable over the last decade (*Pavés et al., 2016*). However, the differences in immune response and hookworm dynamics in the studied population suggest that hookworm biomass and infection duration are the key parasite-specific components that drive mortality if prevalence is relatively stable. In line with these observations, the control of the immune system on the clearance process and the sensitivity of fur seal immune system to external factors such as maternal attendance, establish a pattern where parasite transmission is indirectly linked to environmental change in the ocean.

Global climate change is increasing ocean temperature, particularly in the Southern Hemisphere (*Wijffels et al., 2016*). Our findings highlight that under this scenario infectious diseases could have a more detrimental impact on populations of fur seals and sea lions in the future, but also provide

a foundation for the study of climate change adaptation options for these species (*Hobday et al., 2015*). For instance, treatment of pups to eliminate or decrease parasite burden could be more productive in years with adverse environmental conditions or on otariids with less flexible foraging strategies (*Costa, 2012*). Considering the long-term impact of hookworm disease on the population fitness, host species extinction risk, and the importance of parasite biodiversity is critical before evaluating intervention strategies.

## Conclusions

In Chilean Patagonia, during years with high SST, ocean productivity decreases, forcing adult female fur seals to increase their foraging trip length and decrease their levels of maternal attendance. Pups receiving less maternal care had reduced growth rates and decreased energy budget, impairing the ability of their immune system to mount an effective response against hookworms to expel the parasite from the intestine. These pups, with longer hookworm infection periods, usually die as a consequence of hookworm disease establishing a pattern in which hookworm disease severity and mortality are correlated to indexes of oceanographic environmental conditions such as sea surface temperature. The sensitivity of otariid hookworm disease to ocean temperature and marine productivity presents a scenario where global climate change could increase the extent and severity of a disease present in most fur seal and sea lion populations.

## Materials and methods

### Fur seal health assessments and mortality

From 2012 through 2017 South American fur seal pups were captured by hand every 7–15 d between December 15th and March 10th. At the first capture, pups were marked with a number in the fur using commercial hair decoloring solution. During each capture procedure, standard length, weight, sex, and body condition were recorded. The pup age was calculated based on the peak of parturition for Guafo Island rookery (December 15, *Pavés et al., 2016*) and based on the assessment of rest of placenta (1–3-d-old) and umbilical cord (2–7-d-old) during the first capture. For a subset of pups during 2014 (n = 10), 2015 (n = 20), and 2017 (n = 40), age was exactly known because their parturition was observed and they were marked 24 h later. Blood was drawn from the caudal gluteal vein of pups into EDTA, heparin, and plain (serum) vacutainer tubes. Plain blood tubes were centrifuged within 1–3 h post-collection in the field laboratory to obtain serum, which was preserved at −20°C until later long-term storage (−80°C) or analyses in the mainland laboratory. Plasma was obtained and stored following similar procedures with heparin non-coagulated blood. During each capture procedure, a rectal swab was collected and stored in Sheather's sucrose for later semi-quantitative determination of hookworm egg burden according to standardized methods for this fur seal population (*Seguel et al., 2018*). Hookworm burden of pups found dead was determined by collection, sexing, and counting of all nematodes present in the small intestine and correlated with egg burden through a fecal swab collected during necropsy (*Seguel et al., 2017*; *Seguel et al., 2018*). Using non-coagulated blood, hematocrit, hemoglobin concentration, total red blood cell count (RBC), total white blood cell count (WBC), and differential leukocyte counting were determined for each pup as previously described (*Seguel et al., 2016*). The total serum concentrations of albumin, globulins, cholesterol, glucose, triglycerides, blood urea nitrogen, and creatinine were determined in the mainland laboratory using previously described methods for this population (*Seguel et al., 2016*). All blood and serum (or plasma) measured parameters were obtained for every capture procedure.

Because previous studies indicated a hookworm prevalence close to 100% in this population (*Seguel et al., 2018*), a 'hookworm-free' control group was created in 2017 by treating 60 pups with a subcutaneous injection of ivermectin (300 µg/kg) when they were between 1 and 7 d old. These pups were subjected to the same capture, handling, health assessments, and data acquisition procedures as indicated for the non-treated pups. These pups never presented hookworm eggs in their feces during the duration of the study.

In 2014 (n = 38), 2015 (n = 53), and 2017 (n = 54) marked pups were observed at least once a week during the study period. Pups observed dead were retrieved from the rookery to perform complete necropsies, collect tissue samples for histopathology, and determine the cause of death

according to previously described diagnostic criteria (*Seguel et al., 2011*; *Seguel and Gottdenker, 2017*). The minimal number of pups to capture every year was calculated at the beginning of the reproductive season based on the known recapture rates at Guafo Island (60–80%) and sample size simulations to reach a power of at least 80% (R packages 'pwr' and 'SIMR'). Therefore, hookworm disease outcome (dead vs. survived) was known in these pups and registered in the final data sheet to calculate total hookworm mortality and to fit models to identify the most significant health-related parameters that predicted hookworm mortality.

## Immune challenge experiments

In 2017, a PHA immune challenge experiment was performed in a group of pups when they were approximately 8 wk old (n = 75). For these animals there were enough recaptures (at least four) to measure the average of all health-related parameters, hookworm infection history, and outcome (survival vs. death) at the end of the study period (February–March). The challenge consisted of injection of 0.1 ml of a 1.0 mg * ml$^{-1}$ solution of phytohemagglutinin (PHA) into the interdigital skin of the right posterior flipper (*Vera-Massieu et al., 2015*). The same volume of a saline solution was injected in the same location of the left flipper (control). Swelling was measured in both injection sites 12 h after challenge and a 4-mm punch biopsy was collected from each site (PHA and control) following anesthesia with 5% isoflurane. Biopsy samples were stored in 10% buffered formalin and routinely processed for histopathology and immunohistochemistry for CD3. The number of CD3 + lymphocytes in control and treatment biopsies were counted and the difference between these two recorded and used in statistical analyses. In 2017, another subset of pups of known age (n = 55) was immune-challenged when they were approximately 30-d-old (during the acute hookworm infection phase). Sample size was calculated based on a minimum power of 80% using data collected in a preliminary study in 2016 (differences in skin swelling and T-cell recruitment). Pups were divided into two groups based on level of skin swelling and number of CD3+ lymphocytes detected during examination of biopsy samples. Animals with more than 20 CD3+ lymphocytes per section were considered high responders, whereas pups with less than 16 CD3+ lymphocytes were categorized as low responders. Only one pup was in the middle range (17 cells) and was not included in comparison analyses. At the end of the study period, data on growth rate, maternal attendance, and hookworm infection status were available for these pups.

## Immune tests
### Anti-hookworm ELISAs

A parasite-specific IgG ELISA was developed using whole worm extract as an antigen. Fresh hookworms were collected during necropsies at Guafo Island, washed in PBS, and frozen at −20°C in the field until transport to the mainland laboratory where they were stored at −80°C.

Thawed nematodes were macerated in phosphate buffered saline (PBS) using a glass homogenizer. The macerated nematodes were centrifuged at 15,000 RPM, 4°C for 1 h. Supernatant was collected, filtered, and total protein concentration determined using Bradford, bicinchoninic acid and 'NanoDrop' methods. Extracts were diluted in PBS for a final protein concentration of 1.6 μg/ml. High binding ELISA plates were coated overnight at 4°C using 100 μl per well of diluted (1:100) hookworm extract. Plates were washed with PBS/Tween and 100 μl of sample (fur seal serum) diluted in 5% dry milk/PBS were added to each well and incubated for 5 min at room temperature. A serial dilution of pooled fur seal serum from samples with high absorbance in previous experiments was used to construct a standard curve in each assay (positive controls). Serum from fur seal neonates and animals not exposed to hookworms (fur seal pups from populations without hookworms) were used as negative controls in each test and during standardization experiments. One hundred microliters of diluted (1:15,000) protein A (Vector laboratories, Burlingame, CA, USA) was added to each well and incubated for 5 min at 25°C. Plates were washed three times and 100 μl of TMB was added to each well and incubated for 30 min at room temperature. ELISA reaction was stopped using 100 μl per well of 1.0 N HCl and the plate was read at 450 nm wave length absorbance. The anti-hookworm IgG concentrations were calculated semi-quantitatively by comparing the optic density (OD) of the standard curve with the OD of the samples and reported as arbitrary units (AU). All these reactions were run in duplicate.

A similar procedure was used to standardize a parasite-specific IgE ELISA using a goat IgG anti-dog IgE antibody (Bethyl laboratories, Montgomery, TX, USA) as primary antibody and a rabbit anti-goat IgG as secondary antibody (Bethyl laboratories, Montgomery, TX, USA). However, we were unable to produce OD readings without a significant amount of noise (background staining) and data produced with this test were not further analyzed.

## Special stains and immunohistochemistry

Sections of small intestine from necropsied pups and the skin sections of pups that underwent PHA immune challenges were routinely processed for histopathology and immunolabelled with antibodies against CD3, IBA1, CD79a, CD21, CD127 (c-kit), MUM1, and IL-4. The details of the antibodies used, retrieval, visualization methods, and dilutions are provided in *supplementary file 12*. The number of positive cells was recorded according to standard methods and used in data analyses.

General steps applied to all IHC protocols included deparaffinization of 4-µm tissue sections through immersion in xylene, and rehydration with graduated alcohols, antigen retrieval, quenching of endogenous peroxidase with hydrogen peroxide 3% for 15–20 min, incubation with primary antibody, blocking of nonspecific binding sites with a commercial blocking solution (Power Block, DAKO, Carpinteria, CA, USA), incubation with biotinylated secondary antibody (1:100 dilution, Vector Laboratories, Burlingame, CA) at room temperature for 20–30 min and with horseradish peroxidase-labeled streptavidin for 15 min (Biocare, Chicago, IL). Antigen-antibody complexes were visualized by incubation at room temperature for 5 min with diaminobenzidine (DAB) (Vector Laboratories, Burlingame, CA). Slides were counterstained with hematoxylin, dehydrated, and coverslipped. Tissue sections were observed in an optic microscope and representative sections photographed. The number of cells with positive imunolabelling was counted using the digital images with the use of the counting function of Adobe Photoshop. Sections from small intestine were stained through PAS-Alcian blue reaction to detect the number of goblet cells and amount of mucin produced. Slides were examined and standard sections photographed to calculate the amount of mucin present in the intestine. This number was calculated using Adobe Photoshop selection and calibration tools as the proportion of the total photographed area that stained positive with PAS-Alcian blue.

To detect the site of binding of anti-hookworm IgG in the body of nematodes, formalin-fixed hookworms obtained during necropsies of SAFS pups were routinely processed for histopathology (*Seguel et al., 2017*). Slides were cut at 4 µm, deparaffinized, and antigen retrieval was performed in citrate pH 6.0 for 10 min at 120°C. Blocking of nonspecific binding sites was done by incubation with 10% dry milk/PBS for 20 min and quenching of endogenous peroxidase by incubation with hydrogen peroxidase 3% for 30 min. After three washes with PBS, slides were incubated for 1 h at room temperature with SAFS pup serum that had the highest (strongly positive) or lowest (negative) absorbance during ELISA experiments, diluted (1:50) in 3% bovine serum albumin. After three washes with dilution buffer TWEEN (DBT), slides were incubated with biotinylated protein-A (1:1000 dilution) (Vector laboratories, Burlingame, CA) for 20 min at room temperature to detect IgG. Slides were washed three times with DBT and incubated with streptavidin horseradish peroxidase for 15 min at room temperature. Antigen-antibody complexes were visualized by incubation with DAB for 5 min at room temperature. Slides were counterstained with hematoxylin, dehydrated, and coverslipped.

## Female foraging trip length, maternal attendance, and pup growth rates

In 2007 and 2017, observational studies were conducted in marked SAFS adult females and pup pairs (2007 n = 128, 2017 n = 78). Females were marked with paint spots delivered through paintball guns or by capture procedures using a net. In these procedures females were tagged in the front flippers (Allflex, Dallas, TX, USA) and marked with a number in the back using paint wax to facilitate their observation. Pups were captured, handled, and marked as previously described. Pups and females were observed daily for 1.5 h in the AM and 1.5 h in the PM. If a female was with her pup, this was characterized as a nursing event (regardless of whether the pup was suckling or not), and if a female was not present at the rookery and her pup was alone, the event was characterized as foraging (*Francis et al., 1998*; *Kirkman et al., 2002*; *Trecu et al., 2010*). All female-pup pairs showed high fidelity and allo-sucking events were not observed. The total number of nursing events observed for each pup was corrected by dividing them for the standard number of continuous

observation events to avoid underestimation in pups with shorter observation periods (*e.g.* pups died). Therefore, for data analyses, the corrected number of observation events was used. Adult female foraging trip length was calculated by adding the number of observation periods (12-h intervals) when the female was not present at the rookery and her pup was alone (*Francis et al., 1998*; *Kirkman et al., 2002*; *Trecu et al., 2010*). Only animals with continuous observations were included in the data series (2007 n = 23, 2017 n = 21).

Pup growth rates were obtained through the following equation:

$GR = (W_2 - W_1)/\Delta d$,

where $GR$ = growth rates (g/day),

$W_1$ = weight at first capture,

$W_2$ = weight at last capture,

$\Delta d$ = d between first and last captures.

The minimal number of days used to obtain growth rates was 30 d ($\Delta d \geq 30$) to assume a linear growth rate (*Doidge et al., 1984*).

## Hookworm prevalence, ocean temperature, and primary productivity data

In Austral summers of 2004–2008 (n = 124) and 2012–2017 (n = 154), fresh fur seal pup carcasses were retrieved from the South American fur seal rookery at Guafo Island, Northern Chilean Patagonia (43° 35′ 34.9″ S, 74° 42′ 48.53″ W). Complete necropsies and histopathology were performed on these carcasses as previously described, to determine the cause of death of each pup (*Seguel et al., 2011*; *Seguel and Gottdenker, 2017*). During necropsies all parasites were collected and stored in 5% formalin for later counting. The median of the number of hookworms per pup in a given year was used as a measurement of hookworm burden for that particular season. The yearly prevalence of hookworm infection was calculated as the total number of pups with hookworms at necropsy divided by the total number of pups necropsied during that season.

Sea surface temperature and chlorophyll-a satellite data was retrieved from the NASA earth observation (NEO) website (https://neo.sci.gsfc.nasa.gov). The latitude and longitude to retrieve the chlorophyll-a and temperature data were selected to represent standard points 25 to 50 km west, south, north, and east of the South American fur seal rookery at Guafo Island. This approach was used to represent all the potential foraging areas of fur seals at Guafo Island (Data S8, sea surface temperature).

## Data analyses

### Hookworm mortality models

To identify factors that affected hookworm-induced pup mortality in 2014, 2015, and 2017, generalized linear mixed effect models (GLMM) were fitted using year as random effect (R package 'glmmTMB', *Brooks et al., 2017*). A multimodel selection approach and statistical inference was performed as recommended for data from natural systems (*Burnham et al., 2011*; *Grueber et al., 2011*). Predictors of mortality tested in different models included the average serum or plasmaconcentrations of albumin, globulins, cholesterol, glucose, and triglycerides, average blood hemoglobin concentration, pup growth rate, hookworm infectious period, the average and highest hookworm burden detected in the pup, and the average numbers of peripheral blood lymphocytes, neutrophils, eosinophils, basophils, and macrophages and their interactions. Growth rate and glucose were never fitted in the same models because of high correlation between these variables ($r^2$ = 0.67). The output of each model and graphics of residuals were assessed to check model assumptions, overdispersion (residuals deviance), goodness of fit, and predictors coefficients and standard errors. Multiple models were constructed by adding and deleting predictors and their interactions based on biological predictions and model outputs. The selected fitted models that met quality assessment in terms of fulfillment of assumptions, overdispersion, and fit were later ranked based on second order Akaike's information criteria (AICc). Additionally, Akaike weights and pseudo-R-squared for mixed models (*Nakagawa and Schielzeth, 2013*; *Nakagawa et al., 2017*) were obtained to compare models explanation of the data. Models with a delta AICc <2.0 were considered equally explanatory and were later averaged using the 'model average' function in the multimodel inference 'R software' package 'MuMIn' (*Bartoń, 2017*). Predictor coefficients, standard errors and p-values were assessed and reported in the text and supplementary tables.

## Capture-recapture health assessments

Based on the recapture data, pups were assigned to different phases of the hookworm infection. These included the prepatent period, which corresponded to the phase when a pup had fresh or recently dried umbilical cord and was negative for hookworms at coprological examination; the patent period, which corresponded to the capture when pups had hookworm eggs in their feces; the clearance period, which corresponded to the capture when pups had a significant decline (at least 50%) in their hookworm burden when compared to their previous coprological exam; and the post-clearance period, which corresponded to the captures when pups with previously positive coprological exam had no hookworm eggs in their feces. Pups were also divided into animals that died from hookworm disease (mortality group), animals that cleared hookworms and survived (survived group), and age-matched control pups. These animals never presented patent hookworm infection because they were treated with ivermectin during their first 5 d of life. The immunological parameters obtained through complete blood cell counts (CBCs), serum chemistry, and ELISAs were compared between the different groups and different infection stages through GLMMs ('glmmTMB' R package) using pup group and infection stage as random effects and pup identification number as a nested random effect within a group or infection stage to account for repeated measures (*Bolker et al., 2009*). Additionally, the mean of metabolic parameters and maternal nursing events of pups that cleared and survived hookworm infection were compared with age-matched control pups and pups that died from hookworm disease through GLM with Gaussian or negative binomial distribution according to data distribution.

Pups found dead from non-hookworm related causes were assumed to be at hookworm clearance based on previous clinical data on that particular pup and findings at necropsy. These pups usually died from drowning or trauma. Pups were assumed to die because of hookworm enteritis and bacteremia according to previously established criteria to diagnose this condition (*Seguel et al., 2017*). The number of immune cells and amount of mucin in these two groups were compared through GLMs with Gaussian or negative binomial distribution.

## Models for growth rate and PHA immune challenge response

To determine the best serum chemistry markers (BUN, cholesterol, glucose, albumin, globulins, hemoglobin, and triglycerides) to predict growth rate, several GLMs with negative binomial distribution were fitted, assessed, and reported as previously indicated for models of mortality. Additionally, similar models were fitted and assessed including only the external factors that could affect growth in the pups. The global model included hookworm burden, hookworm infectious period, number of nursing events, sex, and ivermectin treatment as predictors.

To determine the factors that affected the level of T-lymphocytes response in the pups challenged with PHA, GLMs with negative binomial distribution were fitted using hookworm burden, number of leukocytes in peripheral blood (basophils, lymphocytes, etc.), growth rate, scaled body mass, sex, and concentration of glucose, cholesterol, and hemoglobin as predictors of the number of CD3+ cells in skin biopsies in the global model. Model fitting and selection procedures were performed as described for the binomial models for hookworm mortality (multimodel inference approach). Models were ranked based on AICc, with top ranked models (delta AICc >2.0) averaged and the output of the averaged models reported. The differences in health parameters, maternal attendance, growth rate, and metabolic parameters between pups with high CD3 response and low CD3 response were assessed through Gaussian or negative binomial GLMs according to type and distribution of data points.

## Temporal variation on health, maternal attendance, and environmental conditions.

The differences in the mean values of foraging trip length, maternal attendance, and pup growth rate between the 2007 and 2017 breeding seasons were assessed through student's T-test. The differences in the mean values of glucose, cholesterol, basophils, lymphocytes, and parasite-specific IgG for the breeding seasons between 2012 and 2017 were assessed through Kruskal-Wallis tests with posterior Dunn's multiple comparison test.

To assess a potential correlation between SST, chlorophyll-a, and hookworm-related variables in the fur seal population, the average of SST and chlorophyll-a measurements for the months of

December through March (fur seal reproductive season) at the different geographic points assessed were correlated (through Spearman-rho) with hookworm prevalence, mortality, and burden. The month and geographic point with the highest r-square was selected to run linear and polynomial regression models to describe the correlation between these two variables. The best fit model was selected based on the lowest AICc and high $R^2$ values for that particular relationship.

All statistical analyses were performed in 'R' statistical software version 3.3.2 (*R Core Development Team, 2017*) and statistical significance was set at alpha = 0.05 for all tests.

### Data and material availability

All data are available in the manuscript or the supplementary materials.

## Acknowledgements

We appreciate the logistical support of the Chilean Navy, Artisanal fishermen of Quellon (Vessel crews Marimar II and Nautylus V), and the crews of the Chilean Navy lighthouse. We thank Amanda Hooper, Eugene DeRango, Elvira Vergara, Ignacio Silva, Dr. Lorraine Barbosa, Emma Milner, Sian Tarrant, Emily Morris, Suzette Miller, and Piero Becker for dedicated field assistance. We thank Dr. Vanesa Ezenwa for comments and insights in earlier versions of the manuscript.

This work was supported by The Rufford Small Grant Foundation (Grant N 18815–1), Morris Animal Foundation (Grant N D16ZO-413), and the Society for Marine Mammalogy Small Grants in aid awards 2015 and 2016.

## Additional information

### Funding

| Funder | Grant reference number | Author |
| --- | --- | --- |
| Morris Animal Foundation | D16ZO-413 | Mauricio Seguel |
| Society for Marine Mammalogy | Small grants in aid | Mauricio Seguel |
| Rufford Foundation | N 18815–1 | Mauricio Seguel |

The funders had no role in study design, data collection and interpretation, or the decision to submit the work for publication.

### Author contributions

Mauricio Seguel, Conceptualization, Data curation, Formal analysis, Supervision, Funding acquisition, Validation, Investigation, Visualization, Methodology, Writing—original draft, Project administration, Writing—review and editing; Felipe Montalva, Data curation, Investigation, Methodology, Writing—review and editing; Diego Perez-Venegas, Data curation, Formal analysis, Funding acquisition, Investigation, Methodology, Project administration; Josefina Gutiérrez, Data curation, Investigation, Methodology, Project administration; Hector J Paves, Conceptualization, Data curation, Supervision, Funding acquisition, Investigation, Methodology, Project administration; Ananda Müller, Carola Valencia-Soto, Elizabeth Howerth, Data curation, Formal analysis, Investigation, Methodology; Victoria Mendiola, Data curation, Formal analysis, Methodology; Nicole Gottdenker, Conceptualization, Formal analysis, Funding acquisition, Methodology, Writing—original draft, Project administration, Writing—review and editing

### Author ORCIDs

Mauricio Seguel http://orcid.org/0000-0002-0465-236X

### Ethics

Animal experimentation: The experiments described in this manuscript were conducted with approval of the Chilean fisheries service and the University of Georgia animal use committee (IACUC #A2013 11-004-Y3-A0).

Decision letter and Author response
Decision letter https://doi.org/10.7554/eLife.38432.034
Author response https://doi.org/10.7554/eLife.38432.035

## Additional files

### Supplementary files

• Supplementary file 1. Selected binomial generalized linear mixed models for hookworm mortality in South American fur seal (*Arctocephalus australis*) pups.
DOI: https://doi.org/10.7554/eLife.38432.019

• Supplementary file 2. Multimodel coefficient estimates, standard errors (SE), z and p values of predictors of hookworm mortality in South American fur seal pups.
DOI: https://doi.org/10.7554/eLife.38432.020

• Supplementary file 3. External factors affecting South American fur seal pup growth rate.
DOI: https://doi.org/10.7554/eLife.38432.021

• Supplementary file 4. Averaged coefficients, standard errors (SE), z and p values of top ranked models for pup's growth shown in *supplementary file 3*.
DOI: https://doi.org/10.7554/eLife.38432.022

• Supplementary file 5. Negative binomial generalized linear models for CD3 lymphocyte response in South American fur seal pups infected with hookworms.
DOI: https://doi.org/10.7554/eLife.38432.023

• Supplementary file 6. Averaged coefficients, standard errors, z and p values of top ranked models for CD3 lymphocyteresponse shown in *supplementary file 5*.
DOI: https://doi.org/10.7554/eLife.38432.024

• Supplementary file 7. Negative binomial generalized linear mixed models for hookworm infectious period in South American fur seal pups.
DOI: https://doi.org/10.7554/eLife.38432.025

• Supplementary file 8. Averaged coefficients, standard errors, z and p values of predictors for hookworm infectious period based on the top ranked models shown in *supplementary file 7*.
DOI: https://doi.org/10.7554/eLife.38432.026

• Supplementary file 9. Hookworm prevalence, median hookworm burden, hookworm mortality, and mean concentration of chlorophyll-a and sea surface temperature (December) during 10 South American fur seal reproductive seasons at Guafo Island, Southern Chile.
DOI: https://doi.org/10.7554/eLife.38432.027

• Supplementary file 10. Regression models with hookworm prevalence, burden, or mortality as response and sea surface temperature as predictor.
DOI: https://doi.org/10.7554/eLife.38432.028

• Supplementary file 11. Regression models with hookworm prevalence, burden, or mortality as response and chlorophyll-a mean concentration as predictor.
DOI: https://doi.org/10.7554/eLife.38432.029

• Supplementary file 12. Detail of sources, clone, retrieval methods, and dilution of primary antibodies used for immunohistochemistry
DOI: https://doi.org/10.7554/eLife.38432.030

• Supplementary file 13. Data files inventory
DOI: https://doi.org/10.7554/eLife.38432.031

• Transparent reporting form
DOI: https://doi.org/10.7554/eLife.38432.032

### Data availability

All data generated or analysed during this study are included in the manuscript and/or uploaded as supplementary materials.

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
