## [Decision Letter]

Thank you for submitting your article "Immune mediated hookworm clearance and survival of a marine mammal decreases with warmer ocean temperatures" for consideration by *eLife*. Your article has been reviewed by four peer reviewers, and the evaluation has been overseen by Christian Rutz as Reviewing Editor and Ian Baldwin as Senior Editor. The following individuals involved in the review of your submission have agreed to reveal their identity: Urszula Krzych (Reviewer #1); Dan Costa (Reviewer #4).

The reviewers have discussed the reviews with one another and the Reviewing Editor has drafted this decision to help you prepare a revised submission.

1) Collectively, the four reviewers have made a relatively large number of technical comments, and we have decided to append their full reports to this decision letter. During the consultation phase, there were no disagreements between reviewers on technical points, so we would like to ask you to carefully consider their combined feedback in your revision. Briefly, amongst other things, the reviewers: (a) pointed out that the relationship between foraging-trip duration and nursing events could be an artefact, if females shifted attendance ashore more into the night during periods of high temperatures; (b) noted that there is no direct evidence for a connection between attendance and pup glucose levels, since attendance was measured in 2007 and 2017, while glucose levels were only recorded in 2017; (c) cautioned that the fact that lymphocyte counts were higher in pups that survived could simply be a reflection of a developmental stage in immune maturity; (d) requested clarification regarding the PHA immune challenge results and some of the statistical analyses; (e) asked whether faecal counts were confirmed to be negative for 'control' animals treated with ivermectin, given the possibility of reinfection through the milk; (f) raised the possibility to enhance pup survival through targeted intervention; and (g) highlighted that the observed patterns fit models that assess disturbance effects on the foraging ability of marine predators.

Otherwise, there was broad consensus that the presentation of the work requires attention, specifically with regards to the following three points:

2) While there was agreement that the study's broad approach is a key strength, the reviewers felt that the different components could be integrated better.

3) The editors had noted at the initial assessment stage that the reporting of sample sizes should be improved and asked that this be addressed in the full submission. The reviewers have independently picked up on this point again, so clearly more work is required to achieve a satisfactory presentation. Please ensure, for the entire manuscript: (a) that every result stated in the main text, or shown in the figures, is accompanied by an unambiguous statement of the sample size and any other relevant information (such as sampling period and criteria for sub-sampling); and (b) that all results are clearly linked to detailed methodological descriptions in the Materials and methods section. To help with (a) and (b), we suggest you produce a summary data inventory in tabulated form that lists all the datasets collected, with information on methods and sample sizes. Where appropriate, table entries should refer explicitly to results reported in the main text, the figures, and the supplementary data files. Finally, since *eLife* does not impose page/word limits, supplementary text and methods should be integrated into the main body of the paper, for the benefit of the readers.

4) The work needs to be situated better in the existing literature on the topic. The reviewers highlighted the omission of two important edited volumes, by Gentry and Kooyman, (1986) and Trillmich and Ono, (1991), respectively, and suggested several other relevant studies that should be discussed. There was a feeling that claims should be toned down in places, and that earlier work that found similar (or contrasting) results needs to be reviewed more comprehensively. Finally, in some cases, citations seemed out of place, so the referencing should be checked carefully throughout.

*Reviewer #1:*

This paper by F. Montalva documents a very interesting set of events that span many years of studying the South American fur seals in Guafo Islands. The authors provide a correlation between rising sea surface temperature, which causes food shortage and in turn forces mothers into a longer foraging time resulting in a shorter maternal attendance to pups and higher incidence of hookworm infection owing to decreased immune function.

I will address my comments regarding only the immune response aspect of this study. The approaches that the authors are using to evaluate immune responses in fur seal pups with or without hookworm infection at prepatent, patent and cleared infection phases are rather very basic, e.g., enumerating immune cell types, measuring levels of specific IgGs and a limited number of cytokines. But this type of study with fur seals does not actually warrant a more sophisticated immune evaluation, nor are there appropriate reagents to utilize for state-of-the-art for evaluating immune responses in fur seals. The measurements are performed in peripheral blood, mesenteric lymph nodes. In addition, the authors include results from immuno-histopathology cross-section of the intestine, the site of the hookworm infection. The results from experimental probing touch upon innate, T cell and B cell responses and are sufficient to support their correlation (claim) that immune responses are responsible for eliminating hookworm infection and hence increase survival among the recovered pups. Enhanced levels of some of the immune cells were observed both in the peripheral blood as in mesenteric lymph nodes of pups that cleared the infection.

I am not particularly familiar with the PHA (a mitogen) challenge assay, but it seems reasonable and the results support the involvement of T cells during the infectious phase.

I would like to suggest that because this article spans so many different areas of biology, the authors include some explanation as to why they chose to examine these particular cell types, e.g., neutrophils, basophils, mast cells, T cell, B cells in seal pups +/- hookworm infection. For those less familiar with immunology, the function of these particular cell types may be unknown. I do find some explanation for measuring particular cell types in the Supplementary text. Perhaps a more expanded version of this explanation could be moved to the main text of the paper.

*Reviewer #2:*

The manuscript describes how hookworm infections affect the development of pups of the South American fur seal. This study nicely connects marine environmental conditions, maternal foraging behavior and pup hookworm infection, immune response, growth and survival. The authors provide data from the year 2017 that females that were observed nursing the pup more frequently increased pup growth rate. Pups that died were observed most rarely together with the mother. They conclude that under colder SST females spend less time at sea on foraging sojourns thereby providing more nutrition (or more frequently nutrition) to the pup which enables faster clearance of the hookworm infection due to stronger immune reaction thereby reducing pup mortality. The study compares a long series of observation (10 years) and comes to the conclusion that climate change through changes in the marine ecosystem will negatively affect pup survival in this species.

I am concerned that for most statements in the manuscript I could not find sample sizes for the described analyses, but only stats. One has to go to the Excel sheets provided as additional supplementary files to find that information. This I consider unacceptable.

References to the literature are quite unbalanced (many mistakes in the references, for example author lists inconsistent with respect to given names)

Georges and Guinet, 2000 missing "on Amsterdam Island".

I was surprised that the classic book by Gentry and Kooyman (eds) Maternal Strategies on land and at sea. Princeton (1986) was not cited as it is the classic on the effects of marine conditions on maternal attendance patterns.

Similarly, the book by Trillmich and Ono (eds) Pinnipeds and El Niño. Springer Verlag (1991) on the effects of El Niño on attendance behavior and survival of various species of pinnipeds would seem of central relevance to the issue.

Results:

Hookworm dynamics are apparently based on data for 146 pups, but it remains unclear whether these were samples from all study years or just a select period (or spread over all years 2005-2008 and 2012-2017). Hookworm prevalence and mortality in Figure 1B is given on a percent basis, but again no sample sizes are provided. In subsection “Hookworm disease dynamics and mortality in fur seal pups” we hear about a subset of marked pups. So, how can you be sure that the hookworm infections calculated for the other years did not count the same pups multiple times?

The immune characteristics of control, surviving and dying pups are based on pups sampled in 2017. We are not told how "controls" are defined (one can find it in subsection “Data Analyses”). I suppose you mean uninfected pups? Again, sample sizes (n=55? As in the Materials and methods section)?

The same applies to the analysis of the relations between maternal attendance and pup growth and survival in the 2017 cohort.

Subsection “In years with high sea surface temperature there is lower maternal attendance, immunity, and increased hookworm induced mortality”. The relationship between foraging trip duration (how was it measured? On how many females?) and "nursing events" (these should better be called attendances as nursing was not necessarily observed) is based on the years 2007 and 2017. If females shifted attendance ashore more into the night during periods of high temperatures, this relationship could be an artifact.

Subsection “Female foraging trip length, maternal attendance, and pup growth rates”: How many female-pup pairs were observed in 2007 and 2017?

Subsection “Female foraging trip length, maternal attendance, and pup growth rates”: How can you estimate within a time period of 30 days what is the minimal number of days to derive a linear estimate of growth rate? Try to explain more clearly what you did.

Subsection “Fur seal population, ocean temperature and primary productivity data”. Give the number of pups necropsied per year.

Discussion

You have no direct evidence of the connection between attendance and pup glucose levels, since attendance differences were measured for 2007 and 2017, but glucose levels only for 2017. Though your speculation here seems reasonable you should be wording it more careful given the lack of direct evidence.

Francis, Boness and Ochoa-Acuña, (1998); Georges and Guinet, (2000); Arnould, (2001) do not even mention glucose. These papers only communicate data on attendance pattern.

Methods

Subsection “Fur seals health assessments and mortality” If the Seguel et al., standard method for determining hookworm burden is still not published, briefly explain how it works.

Subsection “Fur seals health assessments and mortality”: How many marked pups were observed in 2014, 2015 and 2017?

Subsection “Fur seals health assessments and mortality”: "The number of pups captured each year was calculated based on the known recapture rates at Guafo Island (60% to 80%) and sample size simulations to reach a power of at least 80% (R packages "pwr" and "SIMR")".

I do not understand what you want to say here. Why do you have to calculate the number of pups caught? You should know your sample sizes?

*Reviewer #3:*

This manuscript describes an interesting and timely study. My concerns are three-fold: one, the authors greatly 'oversell' their results and largely ignore other studies that have shown similar (or contrasting) results. Two, some of the key concepts relevant to the study appear to be either treated superficially or not well understood. Three, in some parts, I found the methods used not clear. For instance, it was very difficult to follow whether the same animals used for the PHA challenges were the ones used to quantify IgG, and whether the samples were collected during the challenges. This is important in terms of the discussion of the results. In regards to the statistical analyses, some key results are omitted, and it is my opinion that they could have used a more robust statistical framework to analyze their data and avoid type I errors. Furthermore, I often sensed that this paper is really two studies that they combined in one manuscript but that were not initially related. In particular, the hookworm analysis seems to be a study on its own, and it was difficult to link it to the main story of environmental-related immune alterations.

The major concerns are listed below:

Abstract: The sentence 'Our results provide a mechanistic explanation of how changes in ocean temperature affect immunity and survival of marine mammals' is misleading. If anything, the authors provided evidence that a common and virulent pathogen plays a role in pup mortality when climatic conditions are less than ideal. Extrapolation of this interesting result is unlikely to hold across other species affected by different pathogens. Please rephrase.

Introduction. The sentence 'Regardless, the mechanisms that drive decreased survival during years with low ocean productivity have not been explored beyond assuming that is due to direct mortality because of starvation' oversees previous studies that have looked at this link. For instance, the recent work by Banuet-Martines and others examined immune competence during years with low ocean productivity and reported a glucose-limited mechanism that correlated with lower immune responses and mortality.

Introduction and Discussion section. The use of the word 'reactivity' when speaking of the immune system appears to be misused in the context of the sentence. Immune reactivity relates to specific cellular activities, which are not related to environmental variables but rather to the presence or absence of specific receptors.

Introduction. The statement that among marine mammals infectious diseases are one of the most significant causes of disease of young individuals is erroneous and misleading. Of course, it holds true for some species that have been studied, but there is certainly no evidence to support this statement.

Introduction. The sentence 'These nematodes live in the small intestine where they bite the mucosa to feed on blood, causing substantial tissue damage, anemia and death (10-12), however it is unclear how the host responds to this infection' largely ignores the various studies published on hookworm-related mortality in phylogenetically related species, such as the Northern fur seal and the California sea lion. Please review the literature and rephrase.

subsection “Hookworm disease dynamics and mortality in fur seal pups”. Please be specific (one or two weeks later is very wide and could be relevant, particularly at the age of the pups they studied).

Subsection “Hookworm disease dynamics and mortality in fur seal pups”. Please provide numbers. A 'subset' of pups does not allow a reader to understand the relevance of their findings.

Subsection “Hookworm disease dynamics and mortality in fur seal pups”. The presentation of the statistics is uncommon. I would like to see the percentage of variance explained, and at least some information on homoscedasticity (perhaps as a supplementary table). Additionally, it appears that the authors' grasp on the statistical analyses selected for the study is not very strong (see some of my other comments). For instance, in subsection “Data analyses”, they state that 'the mean of metabolic parameters and maternal nursing events of pups that cleared and survived hookworm infection were compared to the mean of these parameters in age matched control pups and pups that died due to hookworm disease through GLM with Gaussian or negative binomial distribution according to data distribution'. Generalized linear models do not 'compare the means'. Please rephrase or select a proper analysis to challenge the working hypotheses.

Figure 1. The writing is very confusing. The authors graphed 'predicted mortality', but the models appear to use observed mortality as a response variable, not predicted mortality. Please rewrite to ensure clarity.

Subsection “Hookworm clearance is immune-mediated”. The statement that pups that survived experienced an increase in the number of blood lymphocytes is misleading. Lymphocyte counts undergo ontogenetic variations, and their finding does not necessarily imply causation, as the authors appear to intend. Nematode parasites rarely activate lymphocytes, and the fact that lymphocyte counts were higher in the pups that survived could simply be a reflection of a developmental stage in immune maturity. If they had pups of the same age that were not infected, and these pups had lower lymphocyte counts, then the authors' statement would be valid. Furthermore, the authors appear to contradict themselves (see subsection “Maternal attendance affects fur seal pup hookworm clearance”).

Subsection “Hookworm clearance is immune-mediated”. This section appears somewhat unlinked to the study's goals. As it reads, it would seem that this is a separate story. The same thing happens in the Discussion section.

Subsection “Maternal attendance affects fur seal pup hookworm clearance”. This section was somewhat cryptic to me. PHA challenges in pinnipeds induce an initial innate response, with limited infiltration of T cells, and is mostly explained by neutrophil infiltration. According to the authors, they measured swelling and obtained the biopsies 12 hours post challenge, which means that the majority of the response would not be driven by lymphocytes. Furthermore, as written, it appears that the authors propose that having higher T-cell counts have higher maternal attendance, blood glucose, and growth rates, which in any case would be the other way around (higher attendance, blood glucose and growth rates leading to better responses). It is essential that this point is clear, as it is framed within the ecological immunology framework, and is in line with previous findings in both pinnipeds (e.g. Banuet-Martines et al., 2017) and birds (Martin et al., 2004).

Figure 3 legend. Are these really 'predicted values of growth rate'? Also, was the relationship only observed for CD3+ lymphocytes? How about other CD subsets?

Subsection “In years with high sea surface temperature there is lower maternal attendance, immunity, and increased hookworm induced mortality”. 'SAFS females foraging trip length was correlated with the number of nursing events, indicating that the more time females spend at sea less likely is to observe them nursing their pup'. This statement is obvious. It is as saying that 'the less time a mother spends with her pup, the less the pup is seen to be with the mother'.

Subsection “In years with high sea surface temperature there is lower maternal attendance, immunity, and increased hookworm induced mortality”. 'Similarly, the average hookworm infectious period was shorter in years with low SST (GLM, Χ^2^=6.95, df=1, P=0.00036).' This is an important finding that was barely discussed in the appropriate section.

Discussion section. The authors state that they showed that the variations in hookworm disease dynamics are associated with specific changes in the immune response against hookworms. Although I do believe that they showed a mechanistic explanation for how hookworm-related mortality (and morbidity) can vary, they certainly did not study hookworm 'disease dynamics'. This would be a very different study, one that would need a much more thorough ecological framework, which was not carried out here.

Discussion section. As before (see my comments above), the authors are underscoring previous studies to highlight their own results. This is not very professional, and suggests, at the very least, a lack of knowledge on the literature surrounding their study.

Discussion section. I do not understand what the authors wish to communicate with the phrase: 'This suggest that there is a time dependent effect of hookworms on the host, probably associated with host resources depletion, which increases the risk of mortality due to hookworm induced peritonitis'. What evidence of host-resource depletion do the authors have? Also, what evidence is there that (if any) resource depletion leads to hookworm induced peritonitis?

In the same sentence, please include a reference to talk about hookworm induced peritonitis which was first described in another otariid species (Spraker et al.).

Discussion section. Please discuss the results in the context of what has already been published in this regard. A clear link between IgG production and glucose levels in otariid pups during high SST events has already been reported (Martines Banuet et al., 2017), and in the context of the various studies on isotopic signatures of maternal feeding habits during oceanographic alterations.

The title of the subsection reads 'Fur seal population, ocean temperature and primary productivity data', but there is no mention on population censuses.

Subsection “Data analyses”. The authors state that the control animals were those treated with ivermectin during the first five days of life. However, based on what is known of the infectious cycle of hookworms in pinnipeds, pups are re-infected constantly via maternal transmission (in the milk). I would like to know if they did any tests to ascertain that the fecal counts were indeed negative. Otherwise, considering the pups as 'controls' is inadequate.

Subsection “Data analyses”. I am curious to why the authors selected modest t-test or Kruskal-Wallis analysis here. This did not allow for them to consider co-variates that they had already identified.

Subsection “Data analyses”. Model selection based purely on AICc is incomplete. Did the authors compare models statistically? Please provide more information or update the models. I suggest the authors familiarize themselves with model selection criteria. For instance, Kullback's symmetric divergence or deviance based criteria.

*Reviewer #4:*

This is an interesting and important manuscipt. This paper is in the area of disease ecology and is a great example of how disease processes are affected by environmental processes. The manuscript was well written.

My major concern is that the use of SST be clearly defined as an index and not as a causative agent. I think the authors understand that but in a few places, the written is a little imprecise.

---

## [Author Response]

Collectively, the four reviewers have made a relatively large number of technical comments, and we have decided to append their full reports to this decision letter. During the consultation phase, there were no disagreements between reviewers on technical points, so we would like to ask you to carefully consider their combined feedback in your revision. Briefly, amongst other things, the reviewers: (a) pointed out that the relationship between foraging-trip duration and nursing events could be an artefact, if females shifted attendance ashore more into the night during periods of high temperatures; (b) noted that there is no direct evidence for a connection between attendance and pup glucose levels, since attendance was measured in 2007 and 2017, while glucose levels were only recorded in 2017; (c) cautioned that the fact that lymphocyte counts were higher in pups that survived could simply be a reflection of a developmental stage in immune maturity; (d) requested clarification regarding the PHA immune challenge results and some of the statistical analyses; (e) asked whether faecal counts were confirmed to be negative for 'control' animals treated with ivermectin, given the possibility of reinfection through the milk; (f) raised the possibility to enhance pup survival through targeted intervention; and (g) highlighted that the observed patterns fit models that assess disturbance effects on the foraging ability of marine predators.

We appreciate the thoughtful and constructive criticism of all reviewers. Since the editor highlighted the key parts of the study that required revision and/or clarification we reply to these comments now and we will proceed with more specific comments highlighted by each reviewer later.

(a) Potential artifact of the relationship between foraging trips and nursing events.

The reviewers point that in years with higher sea surface temperature (SST), females could shift their attendance more during night, producing an artifact in our data. We cannot completely rule out this possibility, however, given the nature of our data collection we think this is unlikely. As indicated in the Materials and methods section, the calculation of foraging trip length was performed in a 12-hour interval based on the observation hours. Therefore, the only chance to cause such artifact for the calculation of foraging trip length would be that females in years with higher SST arrived to the rookery more often just after the last observation time (after 9:00-10:00 PM). These females would be observed the next morning (7:00-8:00 AM) and therefore their foraging trip length could be increased in up to 11 hours. If this artifact occurred, we would expect to see a higher number of females recorded coming back from their foraging trip in the morning during 2017 compared to 2007. Traditionally, we see more females coming back from foraging trips very early in the morning (one of the reasons for the selection of our observation times, besides making observations easier), however we detected no significant differences in the proportion of observations in the morning in 2007 (78/115, 67.8%) vs 2017 (87/135, 64%).

In the case of nursing events, these could have been more frequent at night hours during 2017, causing fewer observations of this behavior during 2017. However, if pups during 2017 had a higher proportion of nursing events that we did not record compared to 2007 we could expect to see similar or higher growth rate in pups during 2017. According to some of the literature cited in the article and suggested by the reviewers, maternal attendance is one of the most significant factors affecting otariid pups’ growth rate (Costa, 2008, McDonald et al., 2012a), however other factors such as milk quality and energy expenditure due to environmental constraints can play a significant role (McDonald et al. 2012b). Therefore, although we cannot completely assume a direct relationship between maternal attendance and growth, if the other external factors affecting growth remained equal (e.g. pups movements, weather) we could infer that the differences are most likely due to differences in the levels of maternal attendance.

Finally, the main point of this part of the study is that SST is associated with changes in the attendance patterns of SAFS. That is the main point of the comparison between 2007 and 2017 and although it would have been ideal to have attendance data for more years this was not possible due to logistical reasons. Regardless, if the changes observed have the artifact of more nursing events at night, still means that there are differences in the attendance patterns in years with low and high SST. The difference in growth rate of pups also suggest that these changes in attendance could have resulted in differences in growth rates between these years.

These thoughts were incorporated in the results and discussion as follow:

Results section: “SAFS females were observed more frequently arriving to the rookery from foraging trips early in the morning (2007= 78/115, 67.8% returning events in the morning, 2017= 87/135, 64% returning events in the morning).”

Discussion section: “Although, due to the nature of our data acquisition we cannot discard changes in nocturnal maternal attendance patters, the similarity in the proportion of foraging events recorded in the morning in years with high and low SST suggest this is a less likely possibility. Furthermore, the reported changes in attendance coincide with slower growth rates of fur seal pups in a year with high SST compared to a year with low SST, suggesting that there could be a decrease in prey availability for SAFSs in years with higher SST, forcing them to spend more time foraging in the ocean and less time nursing their pups. This interpretation of the results is in line with the proposed conceptual model on the effect of environmental fluctuation in parental attendance in sea lions and fur seals (Costa, 2008). This model suggests that when environmental variation affect prey resources, adult females will increase their foraging intensity effort and metabolic rate before increasing foraging trip length, because the latter almost always result in a decrease in the net energy delivered to the offspring (Costa 2008, 2012).”

(b) No direct evidence of connection between maternal attendance and glucose levels.

Blood glucose levels depend on several factors in mammals, however, the most important in free-ranging animals are food intake and energy expenditure. The argument of the reviewers is that there is no evidence for a link between glucose levels and maternal attendance because attendance was measured only during 2017 and 2007, whereas glucose only in 2017. Unfortunately, we did not collect serum samples in 2007, therefore glucose levels and other metabolites or immune parameters could not be measured during that season. However, we had serum collected from 2012 through 2017, allowing us to measure metabolic and immune parameters these years in order to test the prediction that the mean values of these parameters in the population would change along with the SST. The data showed in Figure 3A and table S2 (current Supplementary Files 3 and 4), suggest that maternal attendance is the most important external factor affecting growth rate of pups in a regular season (2017). Therefore, since data on maternal attendance was not available for the years 2012 through 2016 we tried to provide the closest indicator of energy input in the pups. Growth rate would have been ideal however for 2012 and 2016 we do not have growth rates because there was not enough number of recaptures with the required time span (at least 30 days). Therefore, the idea was try to measure or compare the best “internal” proxy for the pup’s growth. Based on GLMs, the best pup related or metabolic predictors of growth rate was the mean glucose levels and to a lesser extent cholesterol. This is the reason why these values are presented in figure 4C instead of nursing events or growth rate.

An additional reason on why we preferred to report glucose levels instead of growth rates is because in similar studies in otariids, glucose and body mass have been used as a proxy for the energy balance of the animal (Banuet-Martinez et al., 2017). Therefore, to make our results more comparable with these studies we selected this metabolite.

Regarding the link between glucose and maternal attendance, at least during 2017 maternal attendance was the most important external factor that predicted glucose levels. Although these results do not imply cause-effect but association if we assume the current models of energy flow in otariid pups we could infer that pups with more maternal attendance receive more energy and have higher glucose levels. This paragraph of the Discussion section was edited as follow to incorporate the reviewers and editor comments:

“As expected, and as reported in other studies in pinnipeds, pups with higher levels of maternal attendance had higher growth rates (Francis et al., 1998, Georges and Guinet, 2000, Arnould and Hindell, 2001). […] In humans and laboratory animals, glucose metabolism is one of the most important factors driving T-cell activity and production of antigen specific IgG (Mohammed et al., 2012, Palmer et al., 2015).”

(c) Lymphocytes counts higher in pups that survived could reflect immune system maturity.

We think this is very unlikely because the comparisons were performed against age matched controls. In all immunological test the animals age was taken into account because we have previously reported in this population that mean values of leukocytes in blood change as pups get older (Seguel et al., 2016). Therefore, controls were selected based on their estimated age and animals that died were also compared against controls and the values of animals that survived at the same age. We have included additional statements in the Materials and methods section to highlight how age was determined and how comparisons between groups were performed on animals of the same age.

Materials and methods section: “The pup age was calculated based on the peak of parturition for Guafo Island rookery (Paves et al.2016) and based on the assessment of rest of placenta (1-3 days old) and umbilical cord (2-7 days old) during the first capture. For a subset of pups during 2014 (n=10), 2015 (n=20) and 2017 (n=40), age was exactly known because their parturition was observed and they were marked 24 hours later”.

“Pups were also divided in animals that died due to hookworm disease (mortality group), animals that cleared hookworms and survived (survived group) and age-matched control pups”.

(d) Clarification regarding PHA immune challenge and some of the statistical analyses.

We have addressed specific comments regarding PHA immune challenge in the responses to the different reviewers. Here we summarize some of the points explained and modification incorporated in the manuscript.

Phytohemagglutinin (PHA) is a lectin protein that once injected in the skin induces a local inflammatory reaction that includes T-lymphocyte recruitment and proliferation. Other cell types such as macrophages, neutrophils and even eosinophils are also observed in the inflammation site (personal observations and Vera-Massieu et al. 2015 “Induction of an Inflammatory response is context dependent in the California sea lion”). In our study, we used several modifications of the approach described in Banuet-Martinez et al. 2017 in order to measure T-cell response instead of non-specific inflammation or skin swelling. First, based on previous experiments, we took biopsies 12 hours after PHA injection because this was the minimum time on which we detected no differences in the number of T-lymphocytes recruited (we compared samples taken 4, 6, 12 and 24 hours after injection). In the study by Banuet-Martinez et al. and in most of the wildlife literature, researchers measure skin swelling after PHA challenge. By measuring only swelling, is impossible to dissect which particular cellular response was stronger in particular animals. In our study, after collection of skin biopsies we performed histopathology and immunohistochemistry for CD3 in order to label T-cells in the skin biopsies. This procedure was also repeated in control skin biopsies from the same animals where only saline instead of PHA was injected. Later, we counted the number of cells in a previously determined number of fields in samples and controls and the difference between these two was the number recorded for that animal and used in the statistical models. Therefore, we feel this procedure gives a better measurement of T-lymphocyte recruitment and proliferation in response to an inflammatory stimulus than what has been described previously in the wildlife and ecology literature.

The Materials and methods sections has been edited as follows “In 2017, a PHA immune challenge experiment was performed in a group of pups when they were approximately 8-weeks-old (n=75). […] Only one pup was in the middle range (17 cells) and was not included in comparison analyses. At the end of the study period data on growth rate, maternal attendance and hookworm infection status was available for these pups”.

Regarding the statistical analyses, the presentation of models in the supplementary files has been changed to make their interpretation clearer. Additionally, in the Materials and methods section the analyses were separated and restructured to follow the same structure as the presentation of the results. The methods used to construct and select models were also expanded as follows:

“Hookworm mortality models

To identify factors that affected hookworm induced pup mortality in 2014,2015 and 2017, generalized linear mixed effect models (GLMM) were fitted using year as random effect (R package “glmmTMB”, Brooks et al., 2017). […] Models with a delta AICc <2.0 were considered equally explanatory and were later averaged using the “model average” function in the multimodel inference “R software” package “MuMIn” (Barton, 2017). Predictors coefficients, standard errors and p-values were assessed and reported in the text and supplementary tables.”

(e) Were fecal counts confirmed to be negative in control pups given the possibility of reinfection with hookworms through the milk?

Since hookworm prevalence is approximately 100% among pups in this population we created a control, hookworm-free pups by treating them with ivermectin when they were 1-7 days-old. These pups were subjected to the same capture and handling procedures as hookworm-infected pups, therefore fecal sampling and parasitological exams were always performed. None of the treated pups shed hookworm eggs during the study. In the text, Materials and methods section, we have expanded the details on the control groups:

“Because previous studies indicated a hookworm prevalence close to 100% in this population (Seguel et al., 2018), a “hookworm-free” control group was created in 2017 by treating 60 pups with a subcutaneous injection of ivermectin (300 µg/Kg) when they were between 1 and 7 days old. These pups were subjected to the same capture, handling, health assessments and data acquisition procedures as indicated for the non-treated pups. These pups never presented hookworm eggs in their feces during the duration of the study.”

(f) Raised the possibility of enhanced pup survival through targeted intervention

We have incorporated these thoughts in the context of current literature in the last paragraph of the Discussion section:

“Global climate change is increasing ocean temperature, particularly in the Southern Hemisphere (Wijffels et al., 2016). Our findings highlight that under this scenario infectious diseases could have a more detrimental impact on populations of fur seals and sea lions in the future, but also provide the foundation for the study of climate change adaptation options for these species (Hobday et al., 2015). For instance, treatment of pups to eliminate or decrease parasite burden could be more productive in years with adverse environmental conditions or on otariids with less flexible foraging strategies (Costa, 2012). Considering the long term impact of hookworm disease on the population fitness, the host species extinction risk and the importance of parasite biodiversity is critical before evaluating intervention strategies.”

(g) Highlighted that the observed patterns fit models that assess disturbance effects on the foraging ability of marine predators.

We have incorporated these comments in several parts of the Discussion including the following:

“This interpretation of the results is in line with the proposed conceptual model on the effect of environmental fluctuation in parental attendance in sea lions and fur seals (Costa, 2008). This model suggests that when environmental variation affect prey resources, adult females will increase their foraging intensity effort and metabolic rate before increasing foraging trip length, because the latter almost always result in a decrease in the net energy delivered to the offspring (Costa, 2008, 2012).”

Otherwise, there was broad consensus that the presentation of the work requires attention, specifically with regards to the following three points:2) While there was agreement that the study's broad approach is a key strength, the reviewers felt that the different components could be integrated better.

We have edited the first and second part of the result section to make a better link between these sections:

“Therefore, the most important host related factors affecting hookworm mortality were energy balance and immune response against the parasite. The parasite related factors affecting mortality suggested that hookworm clearance, by reducing infectious period and hookworm burden, enhanced host survival.

Hookworm clearance is immune-mediated

In order to know the mechanisms that drive hookworm clearance and affect host mortality, the immune response to hookworms was investigated during 2017 at the different infection stages in 54 fur seal pups and compared to 24 hookworm-free (ivermectin-treated) age matched controls. During the patent [...]”

Additionally, we have incorporated Figure 7 that summarizes the major findings and interpretation of the study. We have included this figure at the end of the summary paragraph of the discussion, however we leave to the editorial team if the figure is better situated at the end of the results, as Supplementary file or as graphical abstract.

3) The editors had noted at the initial assessment stage that the reporting of sample sizes should be improved and asked that this be addressed in the full submission. The reviewers have independently picked up on this point again, so clearly more work is required to achieve a satisfactory presentation. Please ensure, for the entire manuscript: (a) that every result stated in the main text, or shown in the figures, is accompanied by an unambiguous statement of the sample size and any other relevant information (such as sampling period and criteria for sub-sampling); and (b) that all results are clearly linked to detailed methodological descriptions in the Materials and methods section. To help with (a) and (b), we suggest you produce a summary data inventory in tabulated form that lists all the datasets collected, with information on methods and sample sizes. Where appropriate, table entries should refer explicitly to results reported in the main text, the figures, and the supplementary data files. Finally, since eLife does not impose page/word limits, supplementary text and methods should be integrated into the main body of the paper, for the benefit of the readers.

Sample sizes have been incorporated through the text, paying attention to the sampling period and group sizes. Additionally, we constructed the data inventory proposed by the reviewer and all data sets and the inventory are uploaded with this submission.

Supplementary text and Supplementary Materials and methods have been moved to the main manuscript.

4) The work needs to be situated better in the existing literature on the topic. The reviewers highlighted the omission of two important edited volumes, by Gentry and Kooyman, (1986) and Trillmich and Ono, (1991), respectively, and suggested several other relevant studies that should be discussed. There was a feeling that claims should be toned down in places, and that earlier work that found similar (or contrasting) results needs to be reviewed more comprehensively. Finally, in some cases, citations seemed out of place, so the referencing should be checked carefully throughout.

All citations have been reviewed and put in the journal format. The following new references (many of them suggested or highlighted by the reviewers) have been included in the Introduction, Discussion section and Materials and methods sections.

Constable et al., 2014; Costa, 2007; Costa, 2012; Grueber et al., 2011; Hobday, Chambers and Arnould, 2015; McDonald et al., 2012a; McDonald et al., 2012b; Mohammed et al., 2012; Nakagawa and Schielzeth, 2013; Nakagawa, Schielzeth and Johnson, 2017; Stephens et al., 2009; Watson et al., 2016; Wijffels et al., 2016

Reviewer #1:[…] I would like to suggest that because this article spans so many different areas of biology, the authors include some explanation as to why they chose to examine these particular cell types, e.g., neutrophils, basophils, mast cells, T cell, B cells in seal pups +/- hookworm infection. For those less familiar with immunology, the function of these particular cell types may be unknown. I do find some explanation for measuring particular cell types in the Supplementary text. Perhaps a more expanded version of this explanation could be moved to the main text of the paper.

As stayed by the reviewer we used the immunological tools that we were able to validate in this species, using most of the time other animal’s species (e.g. dog) reagents.

We have incorporated the comments regarding rationale for the use of different cells and immunological measurements in the Results section as follows:

“In order to know the mechanisms that drive hookworm clearance and affect host mortality, the immune response to hookworms was investigated during 2017 at different infection stages in 54 fur seal pups, and compared to 24 hookworm-free (ivermectin-treated) age matched controls.”

“The number of peripheral blood leukocytes (lymphocytes, macrophages, neutrophils, eosinophils and basophils) were obtained as a basic tool to indirectly measure the level of proliferation of these different immune cell types in infected and control animals.”

“In order to determine the morphological and immune cell population changes in the anatomical site of hookworm infection, sections of small intestine and mesenteric lymph nodes were collected from pups that died due to hookworm disease (n=21), pups that were undergoing clearance (n=18) and pups that were never infected with hookworms (controls, n=6).”

Reviewer #2:[…] I am concerned that for most statements in the manuscript I could not find sample sizes for the described analyses, but only stats. One has to go to the Excel sheets provided as additional supplementary files to find that information. This I consider unacceptable.References to the literature are quite unbalanced (many mistakes in the references, for example author lists inconsistent with respect to given names)Georges and Guinet, 2000 missing "on Amsterdam Island".I was surprised that the classic book by Gentry and Kooyman (eds) Maternal Strategies on land and at sea. Princeton (1986) was not cited as it is the classic on the effects of marine conditions on maternal attendance patterns.Similarly, the book by Trillmich and Ono (eds) Pinnipeds and El Niño. Springer Verlag (1991) on the effects of El Niño on attendance behavior and survival of various species of pinnipeds would seem of central relevance to the issue.

The mentioned references were not added in the first version to privilege the inclusion of more current literature in the subject. However, the same principles discussed in the literature mentioned by the reviewer, particularly Trillmich et al. are valid and current. We have included Trillmich et al. in the current version of the manuscript.

Regarding sample sizes, we have included the “n=” or “d.f” in each test reported in the text and figure legends.

Results:Hookworm dynamics are apparently based on data for 146 pups, but it remains unclear whether these were samples from all study years or just a select period (or spread over all years 2005-2008 and 2012-2017). Hookworm prevalence and mortality in Figure 1B is given on a percent basis, but again no sample sizes are provided. In subsection “Hookworm disease dynamics and mortality in fur seal pups” we hear about a subset of marked pups. So, how can you be sure that the hookworm infections calculated for the other years did not count the same pups multiple times?The immune characteristics of control, surviving and dying pups are based on pups sampled in 2017. We are not told how "controls" are defined (one can find it on line 539-540). I suppose you mean uninfected pups? Again, sample sizes (n=55? As in the Materials and methods section)?The same applies to the analysis of the relations between maternal attendance and pup growth and survival in the 2017 cohort.

Hookworm infectious period and clearance patterns were measured in 2014, 2015 and 2017. We have included the following numbers in the report of the results “(2014-15 and 2017, mean=25.7 ± 10.9, n=146)”.

Hookworm prevalence was calculated based on animals necropsies from 2004-08 and from 2012-2017 as indicated in Figure 1B. Sample sizes were incorporated in the text of this part of the Results: “Between 81% to 100% of pups examined through necropsy between 2005-2008 (n=124) and 2012-2017 (n=154) had evidence of hookworm infection, and hookworm-related mortality corresponded to 13%-50% of all pups found dead (n=56, Figure 1B”.

Regarding the possibility of counting a pup more than one time for prevalence estimates, this would be very unlikely given that all pups were marked with a number in the fur. In the next season, pups from the previous season are not present in the rookery given the natural history of the studied species. Even if a few pups could remain, these are attaining foraging independence and are morphologically very different from pups of the new season (a difference of more than 15 Kg plus change in fur color).

Regarding control pups we now report the numbers and characteristics of infected and control groups in the Results: “In order to know the mechanisms that drive hookworm clearance and affect host mortality, the immune response to hookworms was investigated during 2017 at the different infection stages in 54 fur seal pups and compared to 24 hookworm-free (ivermectin-treated) age matched controls.”

The sample sizes were also incorporated in the result section regarding maternal attendance and pup growth.

Subsection “In years with high sea surface temperature there is lower maternal attendance, immunity, and increased hookworm induced mortality”. The relationship between foraging trip duration (how was it measured? On how many females?) and "nursing events" (these should better be called attendances as nursing was not necessarily observed) is based on the years 2007 and 2017. If females shifted attendance ashore more into the night during periods of high temperatures, this relationship could be an artifact.

This point was clarified in the response to the main points highlighted by the editor and all reviewers.

Subsection “Female foraging trip length, maternal attendance, and pup growth rates”: How many female-pup pairs were observed in 2007 and 2017?

The following information was added to the sentence “In 2007 and 2017, observational studies were conducted in marked SAFS adult females and pup pairs (2007 n=128, 2017 n=78)”

Subsection “Female foraging trip length, maternal attendance, and pup growth rates”: How can you estimate within a time period of 30 days what is the minimal number of days to derive a linear estimate of growth rate? Try to explain more clearly what you did.

Since growth rate is not linear, we used a minimum number of days between captures (n=30) to calculate the growth rate as a linear function. This is based on similar studies in pinnipeds (see Doidge et al., 1984).

The text was edited as follow to clarify this point “The minimal number of days used to obtain growth rates was 30 days (∆*d*≥30) in order to assume a linear growth rate (Doidge et al., 1984). “

Subsection “Fur seal population, ocean temperature and primary productivity data”. Give the number of pups necropsied per year.

The sentence was edited as follow “In Austral summers of 2004-2008 (n=124) and 2012-2017 (n=154)”.

I preferred to give the number by period since reporting the total number each year will make the reading more confusing.

DiscussionYou have no direct evidence of the connection between attendance and pup glucose levels, since attendance differences were measured for 2007 and 2017, but glucose levels only for 2017. Though your speculation here seems reasonable you should be wording it more careful given the lack of direct evidence.

This paragraph was rewritten as follow to incorporate the comments from this and other reviewers:

“As expected, and as reported in other studies in pinnipeds, pups with higher levels of maternal attendance had higher growth rates (Francis et al., 1998, Georges and Guinet, 2000, Arnould and Hindell, 2001). […] For instance, recent studies have shown that California sea lion pups with better body condition and higher glucose levels have higher total IgG (Banuet-Martinez et al., 2017).”

Francis, Boness and Ochoa-Acuña, (1998); Georges and Guinet, (2000); Arnould, (2001) do not even mention glucose. These papers only communicate data on attendance pattern.

The references have been changed next to the proper statement as indicated in the response to the previous comment.

MethodsSubsection “Fur seals health assessments and mortality” If the Seguel et al., standard method for determining hookworm burden is still not published, briefly explain how it works.

The reference has been changed to “Segue et el., 2018”. This manuscript is now published in an open access journal.

Subsection “Fur seals health assessments and mortality”: How many marked pups were observed in 2014, 2015 and 2017?

The sentence was edited to read “In 2014 (n=38), 2015 (n=53) and 2017 (n=54) marked pups were observed at least once a week during the study period.”

Subsection “Fur seals health assessments and mortality”: "The number of pups captured each year was calculated based on the known recapture rates at Guafo Island (60% to 80%) and sample size simulations to reach a power of at least 80% (R packages "pwr" and "SIMR")".I do not understand what you want to say here. Why do you have to calculate the number of pups caught? You should know your sample sizes?

The sentence refers to the minimum number of animals to capture every year. The sentence was edited to read “The minimal number of pups to capture every year was calculated at the beginning of the reproductive season based on the known recapture rates at Guafo Island (60% to 80%) and sample size simulations to reach a power of at least 80% (R packages “pwr” and “SIMR”)”.

Reviewer #3:[…] The major concerns are listed below:Abstract: The sentence 'Our results provide a mechanistic explanation of how changes in ocean temperature affect immunity and survival of marine mammals' is misleading. If anything, the authors provided evidence that a common and virulent pathogen plays a role in pup mortality when climatic conditions are less than ideal. Extrapolation of this interesting result is unlikely to hold across other species affected by different pathogens. Please rephrase.

The sentence was changed to “We provide a mechanistic explanation regarding how changes in ocean temperature and maternal care affect infectious diseases dynamics in a marine mammal.”

Introduction. The sentence 'Regardless, the mechanisms that drive decreased survival during years with low ocean productivity have not been explored beyond assuming that is due to direct mortality because of starvation' oversees previous studies that have looked at this link. For instance, the recent work by Banuet-Martines and others examined immune competence during years with low ocean productivity and reported a glucose-limited mechanism that correlated with lower immune responses and mortality.

The sentence was changed to “have not been intensely explored beyond assuming that is due to direct mortality because of starvation”.

We do not intent to disregard similar studies, in fact in the following sentences we discuss in detail the findings on the study mentioned by the reviewer (Banuet-Martinez et al., 2017). However, this study does not provide information regarding mortality or survival of the studied species, only provides data related to energy balance and immune competence in years with different SST.

Introduction and Discussion section. The use of the word 'reactivity' when speaking of the immune system appears to be misused in the context of the sentence. Immune reactivity relates to specific cellular activities, which are not related to environmental variables but rather to the presence or absence of specific receptors.

The word “reactivity” was changed for “immune function”.

Introduction. The statement that among marine mammals infectious diseases are one of the most significant causes of disease of young individuals is erroneous and misleading. Of course, it holds true for some species that have been studied, but there is certainly no evidence to support this statement.

The sentence was reworded as “In some marine mammal populations”.

Additionally, the references provided are just a few examples of studies on marine mammals populations where infectious diseases are among the most significant causes of mortality. As in many other animal groups (and humans), particularly those dependent on parental care, infectious diseases are usually more prevalent among young individuals.

Introduction. The sentence 'These nematodes live in the small intestine where they bite the mucosa to feed on blood, causing substantial tissue damage, anemia and death (10-12), however it is unclear how the host responds to this infection' largely ignores the various studies published on hookworm-related mortality in phylogenetically related species, such as the Northern fur seal and the California sea lion. Please review the literature and rephrase.

To the best of our knowledge there are no studies on the immune response or overall host reaction to hookworm infection in otariids. The closest study dealing with these variables is the work by Marcus et al., 2015. However, in the mentioned study, due to the particular reproductive ecology of the studied host species, it was not possible to completely isolate the effect of the parasite. Most of the studies that have measured the effect of hookworms on otariids have focused on the epidemiology and effect on growth rate. Because of the space limitations within the paragraphs and particular sections is not possible to cite all otariid hookworm literature but now we cite the mentioned study by Marcus et al., and two recent reviews (Lyons et al., 2011 and Seguel and Gottdenker, 2017) that summarize the major aspects of hookworm life history traits, diversity and effects on populations of northern fur seals and also in all wildlife species. The latter review (written by the first and last authors of this manuscript) contains an updated list of the literature on otariids hookworms and their major findings.

Subsection “Hookworm disease dynamics and mortality in fur seal pups”. Please be specific (one or two weeks later is very wide and could be relevant, particularly at the age of the pups they studied).

The sentence was changed to Seven to 15 days before […]”

Subsection “Hookworm disease dynamics and mortality in fur seal pups”. Please provide numbers. A 'subset' of pups does not allow a reader to understand the relevance of their findings.

The number of pups included in the analysis in each year was incorporated.

Subsection “Hookworm disease dynamics and mortality in fur seal pups”. The presentation of the statistics is uncommon. I would like to see the percentage of variance explained, and at least some information on homoscedasticity (perhaps as a supplementary table). Additionally, it appears that the authors' grasp on the statistical analyses selected for the study is not very strong (see some of my other comments). For instance, in subsection “Data analyses”, they state that 'the mean of metabolic parameters and maternal nursing events of pups that cleared and survived hookworm infection were compared to the mean of these parameters in age matched control pups and pups that died due to hookworm disease through GLM with Gaussian or negative binomial distribution according to data distribution'. Generalized linear models do not 'compare the means'. Please rephrase or select a proper analysis to challenge the working hypotheses.

The explanation on the statistical tests used, their reasoning, assessment of homogeneity of variance and proper references has been expanded in the Materials and methods sections. We have incorporated the comments from this and other reviewers. One of the reviewers’ suggestions, regarding percentage of variance explained by the models, was incorporated as pseudo-R-squared measurements from GLMM as recommended by Nakagawa and Schielzeth, 2013.

Additionally, we have changed the format of presentation of models output and now each model is divided in 2 tables. One reporting the predictors included in top ranked models along with basic information for interpretation (delta AIC, pseudo R-squares) and a second table reporting the multimodel (averaged) coefficients, SE and P values of predictors included in the top ranked models. This approach will increase the number of Supplementary file but it will make interpretation and assessment of models easier and less confusing.

Regarding the comment on the “mean of metabolic parameters” does not imply that we compared “means” through GLMs but rather that to obtain the value of each parameter of a pup we used the average metabolic values for that individual (each pup had 4 to 8 measurements throughout the study) since many of these are quite variable depending on fasting state (e.g. BUN). However, this last part of the paragraph is redundant since we do not include the results of those analyses and the GLMM approach with nested random effects was preferred and it is what is reported and explained earlier in the same paragraph. Therefore, we have deleted the section highlighted by the reviewer and edited the “Data Analyses” subsection to make easier to relate the description of methods with results presented in the manuscript.

Figure 1. The writing is very confusing. The authors graphed 'predicted mortality', but the models appear to use observed mortality as a response variable, not predicted mortality. Please rewrite to ensure clarity.

The label on Figure 1 was corrected (“Predicted mortality” was changed for “Mortality”). The figures legend was modified as “Predictors of hookworm mortality in generalized linear mixed models (GLMM) (2014, 2015, 2017) vs observed hookworm mortality.”

*Subsection “Hookworm clearance is immune-mediated”. The statement that pups that survived experienced an increase in the number of blood lymphocytes is misleading. Lymphocyte counts undergo ontogenetic variations, and their finding does not necessarily imply causation, as the authors appear to intend. Nematode parasites rarely activate lymphocytes, and the fact that lymphocyte counts were higher in the pups that survived could simply be a reflection of a developmental stage in immune maturity. If they had pups of the same age that were not infected, and these pups had lower lymphocyte counts, then the authors' statement would be valid. Furthermore, the authors appear to contradict themselves (see subsection “*Maternal attendance affects fur seal pup hookworm clearance”).

We clarified this point previously in the response to the general comments highlighted by the editor as critical. In summary, as stated in the Material and methods and later in the Results section these pups were compared to age matched controls. I think there was some confusion related to the reporting of results in this section. First, we clarify that pups that survive increase the number of lymphocytes. As written in the results we do not interpret the finding but just mention it. This is the expected result given the known ontogenic changes observed in the peripheral blood leukocytes of this species (see for instance Seguel et al., 2016). These changes were observed only in pups that survived and controls, but not in pups that died due to hookworm disease. In other words, pups that did not survive infection failed to follow the expected curve of increase in lymphocytes. Later we report the comparison with age matched controls and pups that died due to hookworm infection. Based on these findings we interpret later in the Discussion section. The fact that pups infected had more lymphocytes than age matches controls suggest an effect of hookworm infection or hookworm treatment in the peripheral blood lymphocytes.

We do not think the results are contradictory, but complementary. In this analysis we compare hookworm burden between infected individuals with high and low CD3 response. First, the analysis is not the same as presented in Figure 2. In this case we are measuring a specific lymphocyte subset in tissues. In Figure 2 we measure all lymphocytes in peripheral blood. Additionally, this assay was performed when pups were 30 days old, therefore a subset of pups were already dead or dying due to hookworm disease and could not be included. Other factor affecting the responses is that many animals with high burden maintain a high CD3 lymphocyte and antibody response and survive hookworm infection. In the design of this experiment the main objective was to determine the main factors associated with high and low CD3 lymphocytes response in the pups, including hookworm infection as one of those factors. Therefore, is not surprising that hookworm burden was not significantly different between pups with high and low CD3 response.

Subsection “Hookworm clearance is immune-mediated”. This section appears somewhat unlinked to the study's goals. As it reads, it would seem that this is a separate story. The same thing happens in the Discussion section.

Based on the results of the first section, our main conclusions were that hookworm related variables, pup’s energy balance, and immune response against hookworm impacted mortality due to hookworms across different seasons. The hookworm related factors affecting mortality were burden and infectious period, which are impacted (Figure 1A) by the clearance process. Therefore, in order to understand how mortality is driven we needed to understand the mechanisms behind hookworm clearance.

We have added the following sentences at the end of the first section to make emphasis in this link: “Therefore, the most important host related factors affecting hookworm mortality were energy balance and immune response against the parasite. The parasite related factors affecting mortality suggested that hookworm clearance, through reduced infectious period and hookworm burden enhanced host survival.”

Later, in the beginning of the second section we have added the following sentences to make the necessary link between these two sections:

“In order to know the mechanisms that drive hookworm clearance and affect host mortality, the immune response of fur seal pups to hookworms was investigated at the different infection stages and in hookworm free (ivermectin treated) age matched controls.”

Subsection “Maternal attendance affects fur seal pup hookworm clearance”. This section was somewhat cryptic to me. PHA challenges in pinnipeds induce an initial innate response, with limited infiltration of T cells, and is mostly explained by neutrophil infiltration. According to the authors, they measured swelling and obtained the biopsies 12 hours post challenge, which means that the majority of the response would not be driven by lymphocytes. Furthermore, as written, it appears that the authors propose that having higher T-cell counts have higher maternal attendance, blood glucose, and growth rates, which in any case would be the other way around (higher attendance, blood glucose and growth rates leading to better responses). It is essential that this point is clear, as it is framed within the ecological immunology framework, and is in line with previous findings in both pinnipeds (e.g. Banuet-Martines et al., 2017) and birds (Martin et al., 2004).

Phytohemagglutinin (PHA) is a lectin protein that once injected in the skin induces a local inflammatory reaction that includes T-lymphocyte recruitment and proliferation. Other cell types such as macrophages, neutrophils and even eosinophils are also observed in the inflammation site (personal observations and Vera-Massieu et al., 2015 “Induction of an Inflammatory response is context dependent in the California sea lion”). In our study, we used several modifications of the approach described in Banuet-Martinez et al., 2017 in order to measure T-cell response instead of non-specific inflammation or skin swelling. First, based on previous experiments, we took biopsies 12 hours after PHA injection because this was the minimum time on which we detected no differences in the number of T-lymphocytes recruited (we compared samples taken 4, 6, 12 and 24 hours after infection). In the study by Banuet-Martinez et al. and in most of the wildlife literature, researchers measure skin swelling after PHA challenge. By measuring only swelling, is impossible to dissect which particular cellular response was stronger in particular animals. In our study, after collection of skin biopsies we performed histopathology and immunohistochemistry for CD3 in order to label T-cells in the skin biopsies. This procedure was also repeated in control skin biopsies from the same animals where only saline instead of PHA was injected. Later, we counted the number of cells in a previously determined number of fields in samples and controls and the difference between these two was the number recorded for that animal and used in the statistical models. Therefore, we feel this procedure, gives a better measurement of T-lymphocyte recruitment and proliferation in response to an inflammatory stimulus than what has been described previously in the wildlife and ecology literature.

We have edited the Materials and methods section accordingly to make these points clearer and also we have modified the redaction of the results and discussion to explain better the suggested link between higher maternal attendance -> blood glucose -> growth rates -> immune response -> hookworm clearance, according to the reviewer’ suggestions.

Figure 3 legend. Are these really 'predicted values of growth rate'? Also, was the relationship only observed for CD3+ lymphocytes? How about other CD subsets?

In Figure 3A we show predicted values of Growth Rate as a function of the number of nursing events. In Figure 3C we show the predicted number of CD3 lymphocytes in response to changes in Nursing Growth rate, and hookworm burden.

Regarding the other CD subsets, unfortunately we were not able to perform CD4 and CD8 to differentiate T cells subpopulations, because the antibodies available for the closely related host species (dog) only worked on frozen tissue sections (of dogs) and due to logistical reasons we were not able to preserve fresh tissues from the fur seal pups (because of the isolated area where we worked and limited freezer space). However, we measured other leukocytes including macrophages (with Iba1) and B-lymphocytes (CD21) however we detected no significant differences in the number of these cells and in the case of B-lymphocytes these were very low numbers in the control and PHA treated biopsies.

Subsection “In years with high sea surface temperature there is lower maternal attendance, immunity, and increased hookworm induced mortality”. 'SAFS females foraging trip length was correlated with the number of nursing events, indicating that the more time females spend at sea less likely is to observe them nursing their pup'. This statement is obvious. It is as saying that 'the less time a mother spends with her pup, the less the pup is seen to be with the mother'.

We agree that this statement seems obvious, however we think is important to include it because rule out some potential bias in the recording of the number of nursing events. If the number of nursing events recorded were not actually linked to female presence or absence from the rookery but to observer ability to find the pups or movements of a female within the rookery we would expect to see a milder correlation and lower R^2^. Additionally, given the comment of other reviewers regarding the potential changes in the nocturnal attendance patterns in years with higher SST, the fact that in 2007 and 2017 the relationship between nursing events and foraging trip remained similar suggest that such effect is less likely. This was explained in more detail in the response to the general comments highlighted by the editor.

Subsection “In years with high sea surface temperature there is lower maternal attendance, immunity, and increased hookworm induced mortality”. 'Similarly, the average hookworm infectious period was shorter in years with low SST (GLM, Χ^2^=6.95, df=1, P=0.00036).' This is an important finding that was barely discussed in the appropriate section.

The following sentences have been added to the Discussion section:

“Additionally, since these immune elements drive hookworm permanence in the pups’ intestine (infectious period), in years with lower SST, hookworm infectious period was shorter, suggesting that environmental variables affect not only hookworm immune response but also transmission patterns.”

Discussion section. The authors state that they showed that the variations in hookworm disease dynamics are associated with specific changes in the immune response against hookworms. Although I do believe that they showed a mechanistic explanation for how hookworm-related mortality (and morbidity) can vary, they certainly did not study hookworm 'disease dynamics'. This would be a very different study, one that would need a much more thorough ecological framework, which was not carried out here.

This manuscript is a continuation or complementary to a work recently published (Seguel et al., 2018) and provided to the reviewers as suggestion from the editorial team. In that study hookworm disease dynamics or ecological aspects are investigated in terms of the natural history of the parasite. Regardless of having investigated these aspects in other studies we have changed the wording of the paragraph to better reflect what is shown in this study (“dynamics” was replaced by “morbidity and mortality”).

Discussion section. As before (see my comments above), the authors are underscoring previous studies to highlight their own results. This is not very professional, and suggests, at the very least, a lack of knowledge on the literature surrounding their study.

We do not think we are underscoring previous studies. To the best of our knowledge there are not other studies that have linked environmental variables with specific immune response to a particular disease and mortality in a marine mammal system. In this paragraph we only highlight the most significant findings of the study and we support some of our statements with some of the most classical studies related to those statements (e.g. ocean conditions affect otariid foraging regimes). It is not our intention in this summary paragraph to make an in-depth discussion of the literature. Additionally, in later sections, we cannot cite all the literature related to otariid foraging ecology and we had to select those studies that resembled our approach more closely to make comparisons. If studies use different methodologies or measure different traits is hard to compare results or put them in the context of our study. Something similar occur with the literature related to hookworm disease, immune response and fur seal reproductive ecology. We have even left some of our previous studies out of the discussion because they do not add significant information in the context of this study. We think that try to synthesize information and discuss the most relevant studies is not “unprofessional” or denotes “lack of knowledge” of the literature, all the contrary, this approach requires an in deep analysis of all studies to select those that are technically sound and more correct to compare with our results, especially if the results in the previous literature or the overall predominant information in the literature is different from what we found in our system.

Regardless of this discussion, we have made an effort to provide additional references particularly to recent studies and some of the “classical” studies suggested by reviewers #3 and #4.

Discussion section. I do not understand what the authors wish to communicate with the phrase: 'This suggest that there is a time dependent effect of hookworms on the host, probably associated with host resources depletion, which increases the risk of mortality due to hookworm induced peritonitis'. What evidence of host-resource depletion do the authors have? Also, what evidence is there that (if any) resource depletion leads to hookworm induced peritonitis?In the same sentence, please include a reference to talk about hookworm induced peritonitis which was first described in another otariid species (Spraker et al.).

In the study cited in this paragraph (Seguel et al., 2017) we have previously explored the potential causes of hookworm peritonitis. We observed that along with hookworm burden, the number of red blood cells in the hookworm intestine was a significant predictor of hookworm peritoneal penetration and peritonitis. Therefore, in that manuscript we propose that the availability of resources for the hookworm could be one of the factors that lead hookworms to dig deeper in the intestinal mucosa and penetrate the intestinal wall. Although additional studies are necessary to test this hypothesis, our other study (Seguel et al., 2018) also shows that hookworms in fur seal pups do not experience density dependent restriction in blood feeding in the intestine, supporting the hypothesis that fur seal hookworms extract host resources in a high rate.

We have edited this sentence as follow to clarify this point:

“This suggest that there is a time dependent effect of hookworms on the host, probably associated with host resources depletion, which could increase the risk of mortality due to hookworm induced anemia and peritonitis (Lyons et al., 2011b, Seguel et al., 2017, 2018).”

Regarding the comment about the first description of hookworm peritoneal penetration in otariids, this was provided by Dr Terry Spraker in a report published in 2004. That study is a case series report on a few individuals. Later in 2007, Dr Spraker et al. published a classical study on the pathology of hookworm infection in sea lions, including description of the prevalence of peritoneal penetration. Since this finding has always been considered relevant in the context of hookworm infection in other species, Dr Lyons et al. published the following study Lyons et al., 2011. In this study, the authors test the hypothesis that infection with the “wrong” parasite species lead to peritoneal penetration. The authors finally reject this hypothesis in the light of their findings, but interestingly found that hookworms in the peritoneal cavity were smaller than those in the intestine, leading to the test in our mentioned study (Seguel et al., 2017) of the “worm starvation” hypothesis. Therefore, I have included Lyons et al. reference instead of Spraker et al. However, I have to caution that the authors of that study (Lyons et al.) do not interpret their data in the context of availability of host resources for the parasite.

Discussion section. Please discuss the results in the context of what has already been published in this regard. A clear link between IgG production and glucose levels in otariid pups during high SST events has already been reported (Martines Banuet et al., 2017), and in the context of the various studies on isotopic signatures of maternal feeding habits during oceanographic alterations.

The following sentences were added to the discussion:

“More recent studies have shown that California sea lion pups with better body condition and higher glucose levels have higher total IgG (Banuet-Martinez et al., 2017). In humans and laboratory animals, glucose metabolism is one of the most important factors driving T-cell activity and production of antigen specific IgG (Mohammed et al., 2012, Palmer et al., 2015).”

The studies on changes in isotopic signatures are discussed later in the section on effects of SST on otariids foraging patterns and health.

The title of the subsection reads 'Fur seal population, ocean temperature and primary productivity data', but there is no mention on population censuses.

The title was changed to “Hookworm prevalence, ocean temperature and primary productivity data”.

Subsection “Data analyses”. The authors state that the control animals were those treated with ivermectin during the first five days of life. However, based on what is known of the infectious cycle of hookworms in pinnipeds, pups are re-infected constantly via maternal transmission (in the milk). I would like to know if they did any tests to ascertain that the fecal counts were indeed negative. Otherwise, considering the pups as 'controls' is inadequate.

There are few studies on the epidemiology and life cycle of hookworms in pinnipeds (see the review by Lyons et al., 2011 and our review on hookworms of wildlife, Seguel and Gottdenker, 2017. Additionally, studies by Castinel et al., 2017 and Marcus et al., 2015). However, all these studies suggest that if reinfection occurs this is not common or extended through time. Additionally, our description of the life cycle of hookworms in South American fur seals suggest that patent infection only occurs through colostrum during the first 1-7 days of the pups life (Seguel et al., 2018).

Regarding the management of control pups, these animals were subjected to the same clinical procedures as infected pups, which included coprological analyses to detect hookworm eggs. All ivermectin-treated pups (controls) never shed hookworm eggs during the duration of the study.

The following paragraph was added to the Materials and methods section.

“Because previous studies indicated a hookworm prevalence close to 100% in this population (Seguel et al., 2018), a “hookworm-free” control group was created in 2017 by treating 60 pups with a subcutaneous injection of ivermectin (300 µg/Kg) when they were between 1 and 7 days old. These pups were subjected to the same capture, handling, health assessments and data acquisition procedures as indicated for the non-treated pups. These pups never presented hookworm eggs in their feces during the duration of the study.”

Subsection “Data analyses”. I am curious to why the authors selected modest t-test or Kruskal-Wallis analysis here. This did not allow for them to consider co-variates that they had already identified.

The purpose of this analysis was only to compare the means of several health and attendance parameters during different years. We did not attempt to examine co-variates since this was performed in earlier sections of the manuscript. Based on the type of data and the distribution and dispersion of the data we considered that these conservative tests were enough to test significant differences between years.

Subsection “Data analyses”. Model selection based purely on AICc is incomplete. Did the authors compare models statistically? Please provide more information or update the models. I suggest the authors familiarize themselves with model selection criteria. For instance, Kullback's symmetric divergence or deviance based criteria.

A multimodel inference approach was used and model selection was based on information criteria but also hypothesis testing throughout the model fitting process as described in Burnham et al., 2011. The following sentences have been added to the methods section to clarify the model fitting and selection procedures employed.

“A multimodel selection approach and statistical inference was performed as recommended for ecological data (Burnham et al., 2011, Grueber et al., 2011).”

“The output of each model and graphics of residuals were assessed to check models assumptions, overdispersion (residuals deviance), goodness of fit and predictors coefficients and standard errors. Multiple models were constructed by adding and deleting predictors and their interactions based on biological predictions and models outputs. The selected fitted models that met quality assessment in terms of fulfillment of assumptions, overdispersion and fit were later ranked based on second order Akaike’s information criteria (AICc). Additionally, Akaike weights and pseudo-R-squared for mixed models (Nakagawa and Schielzeth 2013, Nakagawa et al. 2017) were obtained to compare models explanation of the data. Models with a delta AICc <2.0 were considered equally explanatory and were later averaged using the “model average” function in the multimodel inference “R software” package “MuMIn” (Barton 2017). Predictors coefficients, standard errors and p-values were assessed and reported in the text and supplementary tables.”